# Amorphous Solid Dispersions (ASDs): The Influence of Material Properties, Manufacturing Processes and Analytical Technologies in Drug Product Development

**DOI:** 10.3390/pharmaceutics13101682

**Published:** 2021-10-14

**Authors:** Raman Iyer, Vesna Petrovska Jovanovska, Katja Berginc, Miha Jaklič, Flavio Fabiani, Cornelius Harlacher, Tilen Huzjak, Manuel Vicente Sanchez-Felix

**Affiliations:** 1Technical Research and Development, c/o Global Drug Development, Novartis Pharmaceuticals Corp., One Health Plaza, East Hanover, NJ 07936, USA; 2Product Development, Lek Pharmaceuticals d.d., Verovškova 57, 1526 Ljubljana, Slovenia; vesna.petrovska_jovanovska@sandoz.com (V.P.J.); katja.berginc@sandoz.com (K.B.); miha.jaklic@sandoz.com (M.J.); tilen.huzjak_ext@sandoz.com (T.H.); 3Technical Research and Development, c/o Global Drug Development, Novartis Pharma AG, Lichtstrasse 35, CH-4056 Basel, Switzerland; flavio.fabiani@novartis.com (F.F.); cornelius.harlacher@novartis.com (C.H.); 4Novartis Institute for BioMedical Research, Cambridge, MA 02139, USA

**Keywords:** amorphous solid dispersions, characterization, manufacturing processes, novel stabilizing polymers, material properties

## Abstract

Poorly water-soluble drugs pose a significant challenge to developability due to poor oral absorption leading to poor bioavailability. Several approaches exist that improve the oral absorption of such compounds by enhancing the aqueous solubility and/or dissolution rate of the drug. These include chemical modifications such as salts, co-crystals or prodrugs and physical modifications such as complexation, nanocrystals or conversion to amorphous form. Among these formulation strategies, the conversion to amorphous form has been successfully deployed across the pharmaceutical industry, accounting for approximately 30% of the marketed products that require solubility enhancement and making it the most frequently used technology from 2000 to 2020. This article discusses the underlying scientific theory and influence of the active compound, the material properties and manufacturing processes on the selection and design of amorphous solid dispersion (ASD) products as marketed products. Recent advances in the analytical tools to characterize ASDs stability and ability to be processed into suitable, patient-centric dosage forms are also described. The unmet need and regulatory path for the development of novel ASD polymers is finally discussed, including a description of the experimental data that can be used to establish if a new polymer offers sufficient differentiation from the established polymers to warrant advancement.

## 1. Introduction

The oral route of drug administration is regarded as the most preferred route for medicines, with more than 85% of drugs sold around the world being administered orally. In this context, the properties of a drug molecule that govern oral absorption are critical to its development. The Biopharmaceutics Classification System (BCS) serves as a guide to predict oral absorption based on the aqueous solubility and permeability of a drug [1,2]. Poor solubility is among the primary causes of low bioavailability for orally administered drugs. Drugs that are slightly soluble to practically insoluble exhibit solubility of ≤0.01% based on the description in the United States Pharmacopoeia (USP) [3]. In a comparison of solubility of 200 oral drugs of various origin as seen in Figure 1, 40–45% were very slightly soluble to practically insoluble, representing 33% of drugs listed in the US Pharmacopeia and 75% of compounds under development and 90% of new chemical entities were regarded as poorly soluble [4,5,6,7,8,9,10,11]. The improvement of solubility is therefore regarded as a key driver for greater bioavailability.

The Noyes–Whitney equation [12] relates mass transfer to the concentration gradient as
(1)$$\frac{dM}{dt}=\frac{DA\left[C_{s}-C_{t}\right]}{h}$$
where *D* is the diffusion coefficient (cm^2^/s), *A* is the cross-sectional area, *h* is the thickness of the hydrodynamic diffusion layer and *C_s_* is the solubility or maximum concentration. Under infinite dilution (sink), the concentration gradient approximates to solubility *C_s_*, resulting in
(2)$$\frac{dM}{dt}=\frac{DAC_{s}}{h}$$

For poorly soluble drugs, increasing aqueous solubility and the surface area are primary means of increasing the rate and extent of dissolution since parameters *D* and *h* are a function of extrinsic factors such as viscosity of dissolution medium and stirring rate. The approaches to improve dissolution rate may be broadly classified as physical and chemical as shown in Table 1.

Among these approaches, the conversion of drugs into an amorphous solid dispersion (ASD) form has gained widespread attention over the last few decades. The ASD of a drug molecularly dispersed in a polymeric matrix has been extensively utilized to improve solubility and bioavailability of poorly soluble drugs [19,25,26,27]. An ASD of vemurafenib (Zelboraf^®^) increased human bioavailability by about five-fold compared to the crystalline form [19]. However, since amorphous forms are thermodynamically unstable, the materials and technologies that enable ASD formation, the subsequent dosage form and the methods of characterization of these systems play a critical role in defining the quality, stability, processability and in-vivo performance of the ASD. There are over forty successfully launched ASD-based drug products in the market that point to an industrial relevance and increasing maturity and robustness of the ASD approach as seen from Figure 2. In this paper, the authors discuss the various aspects associated with development of ASDs from a molecule to a medicine including challenges associated with transfer from a laboratory set-up to commercial manufacturing and the need for novel polymers that enable ASD-based medicinal dosage forms.

## 2. Theoretical Considerations

Over the last few decades, there have been several papers published contributing significantly to the body of knowledge and understanding of ASDs [8,28,29,30,31,32,33,34,35,36,37,38]. Some of the underlying theoretical concepts involved in formation and characterization of ASD are presented here.

The amorphous form of a material is obtained by cooling a highly viscous liquid fast enough to avoid nucleation and growth of crystals or crystallization, [29] resulting in a supercooled liquid. This process is commonly referred to as quenching and is illustrated in Figure 3. As the cooling progresses, the amorphous form is trapped or “frozen” due to the continuously increasing viscosity (continuous hardening) of the rapidly cooling melt. The amorphous form obtained in this manner is a super-cooled solid or glass, a disordered material in a metastable state that lacks the periodicity of crystals, but behaves mechanically, similar to a solid [29,30]. In contrast, crystallization of the melt occurs with an abrupt change in viscosity (discontinuous hardening), during which the crystallites grow in the body of the melt. Since the thermodynamic driving force for any transition such as phase or glass transition is the lowering of Gibbs free energy (∆G_f_ = ∆H_f_ − T∆S_f_), the rate of energy loss determines the attributes of the glass [31].

The quenching can be achieved via temperature modulation (e.g., melt quenching, cold precipitation) or by adiabatic expansion (e.g., spray drying, freeze-drying, supercritical CO_2_, etc.). The glass remains in a state of metastable thermodynamic equilibrium as a function of its temperature and the quenching rate.

### 2.1. Temperatures in the Supercooling of Amorphous Solid

The events during the formation and stabilization of amorphous form are characterized by transitions in temperatures as outlined below.

#### 2.1.1. Melting Temperature, T_m_

A first-order, phase transition of solid-to-liquid (fusion) or liquid-to-solid (crystallization) occurs at the melting temperature, T_m_, characterized by ΔG = 0 (free energy change) and ΔH_m_ = enthalpy of fusion or freezing at T_m_. The heat capacity, C_p_, exhibits a discontinuity at the T_m_ since the liquid phase has a larger heat capacity than its crystalline solid [39]. During dissolution of a crystalline solid at a temperature, T (e.g., storage temperature), the free energy change is estimated from its melting temperature and enthalpy of melting using the Hoffmann equation [40]
(3)$$\Delta G=-{\Delta H}_{m}\;\left[T_{m}-T\right]T/T_{m}^{2}$$

The solubility, *S*, of amorphous form over crystal is predicted as a first-order function
(4)$$\ln(S_{a}/S_{c})=\Delta G_{\;}/RT_{m}$$
where *R* is the universal gas constant and the suffixes *a* and *c* represent amorphous and crystal form, respectively, although experimentally determined values have been observed to be lower than predicted [41].

#### 2.1.2. Glass Transition Temperature, T_g_

Quenching at fast cooling rates causes a decrease in volume (increase in viscosity) resulting in the molecules rearranging so slowly that they eventually remain “frozen” in some metastable configuration. The transition to this fixed but metastable configuration occurs at a temperature known as the glass transition temperature, T_g_ as shown in Figure 4.

At a thermodynamic level, T_g_ is the temperature at which an abrupt shift occurs in volume or enthalpy as a function of temperature, ∂V/∂T or ∂H/∂T [29,42]. At a mechanical level, it is the temperature at which the viscosity of the liquid during quenching reaches a large value on the order of 10^12^ Pa·s [43]. At a molecular level, it is the temperature at which the rate of cooling is faster than the rate of relaxation such that the relaxation reaches a fixed value, less than what is needed for crystallization [8]. While melting is a first-order transition at fixed T_m_, T_g_ is a second-order transition with a kinetic component (relaxation) since the rate of change of state variables such as enthalpy and entropy is slower after T_g_. For most drugs, T_g_ occurs at about 0.66 T_m_ [8].

#### 2.1.3. Kauzmann Temperature, T_K_

If the ∂V/∂T line for the cooling liquid in Figure 4 is linearly extrapolated without structural relaxation at T_g_ and the system remains at equilibrium throughout the quenching process, it eventually intersects the crystal line. The temperature at the point of intersection as shown in Figure 4 is called the Kauzmann temperature [44]. If continued further to 0 K (T), the line would intersect at a point where the entropy of the glass is less than that of the crystal, violating the third law of thermodynamics of ΔS of a pure crystalline solid being zero at absolute zero. This is also referred to as the “Kauzmann paradox” or “entropy crisis”. This crisis is avoided when the system conserves the excess entropy at the T_g_ and loses it over time as part of the enthalpic relaxation of the glass. At T_K_ = T_g_, the glass formed has the lowest energy (ideal glass) with zero entropy. Exceptions to the “entropy crisis” due to lack of change in specific volume between liquid and crystal phase (ΔV = 0) have been reported [45]. The Kauzmann temperature is determined as [29]
ΔS_m_ = ΔC_p_ ln (T_m_/T_K_)(5)
ΔS_m_ (melting entropy) and ΔC_p_ are the difference in values of entropy and heat capacity between the liquid and crystal at the melting temperature, T_m_.

#### 2.1.4. Fictive Temperature, T_F_

The fictive temperature, T_F_, of a glass is a measure of its configurational entropy (discussed later) and relates to the kinetic component of relaxation when a glass loses its excess entropy when aged over time (annealing) and reaches a state of equilibrium [43]. The fictive temperature is used to express the temperature and time dependence of molecular mobility. In enthalpic relaxation during annealing, T_F_ (t = 0) and T_F_ (t = duration of annealing) are considered to represent fresh and annealed glass, respectively [46]. T_F_ (t = 0) is quantitatively similar to T_g_ when heat capacity of glass and liquid are similar and T_F_ (t) = T_0_ when heat capacities of glass and crystal solid are the same.

### 2.2. Molecular Mobility, Structural Relaxation and Relaxation Time

A system in non-equilibrium undergoes spontaneous transition (dynamic relaxation) to a state of equilibrium via molecular rearrangements (molecular relaxation) that involves both thermodynamic (loss of free energy, enthalpy and entropy) and physical changes (viscosity, volume). In glass formation, relaxation begins at quenching or supercooling phase of the liquid. As the cooling liquid nears T_g_, its viscosity increases to about 10^12^ Pa·s [29,43] reducing its molecular mobility and short relaxation time (time for molecular rearrangement) in the order of a few hundred seconds. At the T_g_, the increase in viscosity results in a “structural arrest” of the configuration followed by a “structural relaxation” during physical aging of the glass when stored at temperatures below its T_g_, in a process called annealing. Relaxation continues until the glass reaches a fully relaxed state below T_g_ [30]. Fresh glass formed by rapidly cooling the melt has much higher free energy, enthalpy and entropy than the fully relaxed, ideal glass. In short, the structural relaxation enables the glass to reach a fully relaxed meta-stable state without transitioning into crystallization, at a rate governed by the temperature and time during cooling (rate of cooling, temperature and time of annealing) [47].

Structural relaxation is accompanied by loss of enthalpy (exothermic) measured reheating the aged or annealed glass to its T_g_ (endothermic) [48]. This enthalpy is related to the degree of relaxation which in turn is a function of molecular mobility. Molecular mobility is thus an indicator of relaxation with lower mobility leading to shorter relaxation times and therefore determines the strength of the glass [8,30]. Mobility within an amorphous system decreases sufficiently enough to provide acceptable physical stability without recrystallization when stored at a temperature 50 °C below the T_g_ [49].

If relaxation begins at T_m_ and completes at T_0_, (T_0_ < T_m_), the total enthalpy lost during this relaxation is expressed as below, where ΔC_p_ is the heat capacity difference between the solid and glass. In Equation (6), the first term ΔH_m_ represents fusion while the second term represents relaxation enthalpy from T_m_ to T_0_.
(6)$$\Delta H_{total}=-\Delta H_{m}+\overset{T_{o}\;}{\underset{T_{m}}{\int }}\Delta C_{p}$$

The relaxation time is determined using the Kohlraush–Williams–Watts stretched exponential function (β) [29] describing general change in a property, x over time due to an input stress as
(x_t_ − x_∞_)/(x_0_ − x_∞_) = exp [−(t/τ)^β^]  (0 < β ≤ 1)(7)
where x_0_, x_t_ and x_∞_ are the property values of x at time zero, t and at maximum decay. In enthalpic relaxation, ΔH_t_ is the enthalpy recovered at temperature, T for a given storage or annealing time, t and ΔH_∞_ is the maximum enthalpy recovered at infinite time. Therefore
ΔH_t_/ΔH_∞_ = exp [−(t/τ)^β^](8)

Equation (8) is the Kohlraush–Williams–Watts (KWW) equation. When β = 1, the relaxation follows Arrhenius behavior. The enthalpy lost in relaxation is also known as configurational enthalpy since it is a measure of the loss of configurations that the system suffers due to the rapid quenching.

### 2.3. Fragility

Fragility is a measure of the temperature dependence of the molecular motions near the (T_g_) [50] and represents the rate and extent of structural relaxation of glass as the viscosity changes when the system approaches T_g_. This determines the strength of the glass, i.e., whether the liquid forms a “strong” or “fragile” glass [29]. Since crystal solid represents zero molecular motion, the simplest measure of fragility is the ratio m = T_g_/T_m_. Based on a compilation of m values for various drugs, the approximation of m = 0.66 (2/3rd rule) is widely used to predict T_g_ from T_m_, with m < 0.66 indicating greater fragility or poor glass forming ability [51]. The Angell plot of log (η) as a function of T_g_-normalized inverse temperature function, T_g_/T is used to determine a fragility index, m [50,52].
(9)$$m=\frac{d\left(\log_{10}\eta \right)}{d\left(\frac{T_{g}}{T}\right)}$$

The width of the T_g_ (onset to offset) has been used to calculate the activation energy, ΔH and the fragility index, m as [50].

at
T = T_g_, m = ΔH/(2.303 R·T_g_)(10)
where ΔH is the enthalpy of molecular relaxation at T_g_ (or activation energy barrier for viscous flow). For large values of m indicative of fragile glass such as sorbitol, an order of magnitude increase in molecular mobility is observed for every 10 K increase in temperature, while strong glass with lower values of m such as zinc chloride exhibit such an increase for every 25 K increase in temperature [50,53]. An alternate measure of fragility is the constant D in the AGV equation also known as Angell strength parameter and is indicative of the difference between T_0_ (temperature at which S_c_ = 0) and T_g_ with high values of D resulting in a greater difference, indicative of a stronger glass. Most drugs are considered fragile with D < 10 or moderately fragile D = 10–30 [8,54].

### 2.4. Theories of Glass Formation

The Vogel–Tammann–Fulcher (VTF) equation relating temperature and viscosity in glass-forming liquids first proposed in the 1920s in its original form is
η = A exp[B/(T − T_0_)](11)
where A and B are temperature-independent constants and T_0_ is an equilibrium temperature, similar to the Kauzmann temperature, T_K_, at which relaxation is complete with negligible molecular mobility and viscosity of the supercooled liquid reaches the highest value (structural arrest). For liquids that form a “strong” glass with small changes in heat capacity, the viscosity was found to exhibit an Arrhenius type relationship against a T_g_-scaled temperature parameter T_g_/T, i.e., log η ∝ T_g_/T in the region 0 < T_g_/T ≤ 1 as shown in Figure 5. with the slope representing the energy barrier for viscous flow [43]. Liquids that form a fragile glass were found to exhibit significant deviations from Arrhenius behavior or a super-Arrhenius behavior with large increases in heat capacity leading to rapid rates of relaxation and a collapse of liquid structure with decreases in temperature as illustrated shown in Figure 5 [29,55].

The Gibbs–DiMarzio and Adam–Gibbs theory of continuum thermodynamics [42,56] was later used to explain the non-Arrhenius behavior of fragile glass using the concept of configurational entropy as a function of time and temperature.

The entropy in a body is a sum of its vibrational and configurational states: S_total_ = S_v_ + S_c_ [57]. S_v_ is vibrational or thermal entropy associated with atomic vibrations or number of thermal states among the atoms. It increases with temperature and leads to polymorphic transitions within a crystal. S_c_ is configurational entropy associated with number of ways in which a set of atoms or molecules can be distributed in space when mixed together as illustrated in Figure 6.

In the Gibbs–DiMarzio and Adam–Gibbs theories, the liquid in the supercooled region near T_g_ goes through multiple configurations or molecular arrangements in regions or subsystems that can cooperatively rearrange (known as CRR) to achieve equilibrium at a given temperature. The entropy of the whole system associated with such configurations is termed configurational entropy, S_c_ or S_conf._ [43,58], a density function distributed against temperature. The S_c_ of the smallest subsystem is denoted as S_c_ or S_conf_. The decrease in S_c_ occurs as the system has fewer configurations available to rearrange [29] and this decrease results in an increase in viscosity, the rate of increase related to the rate of disappearance of S_c_ [52,59]. In this manner, thermodynamic equilibrium is achieved at each configuration throughout cooling until a second-order transition occurs at T_0_, the temperature at which S_c_ = 0 (or lowest value) forming ideal glass [31] and remains so at T < T_0_. Greater the excess S_c_ at T_g_, slower the relaxation (over a longer time) and stronger the glass. The Gibbs–DiMarczio approach thus resolves the “entropy crisis” with S_c_ = 0 at T_0_ ≈ T_K_. The Adam–Gibbs (AG) equation relating relaxation time (kinetics) to configurational entropy (thermodynamics) is expressed as
τ = τ_o_ exp[B/(TS_c_)](12)
where τ is the molecular relaxation time at temperature T, τ_o_ is the shortest molecular relaxation time, B is a constant = Δμ·s_c_/R and Δμ is the free energy barrier hindering molecular rearrangement, a measure of relaxation enthalpy.

An ideal glass formed at the T_g_ with no change in S_c_ (S_c_ = C_p/_T_g_) is indicative of a “strong” glass with an Arrhenius type relaxation in the supercooling phase up to T_g_, i.e., log τ α T_g_/T. In case of annealing, τ (T, T_f_) and S_c_ is a function of the configurational heat capacity ΔC_p_ where
(13)$$S_{c}=S_{c}\left(T_{o}\right)+\int_{T_{o}}^{T_{f}}\frac{\Delta C_{p}}{T}dt$$

The AG expression becomes
τ = τ_o_ exp [D·T_0_/(T(1 − (T_0_/T_F_)))](14)
where D is the strength parameter, a measure of glass fragility (discussed later).

Equation (11) is known as the Adam–Gibbs–Vogel (AGV) equation [58] used to describe the relaxation behavior of glass when the structure is “frozen in” at T_g_ to its fully relaxed state via annealing. In the absence of annealing, S_c_ = ΔC_p_ [(1/T_0_) − (1/T)], the difference in heat capacity between supercooled liquid and stable crystalline solid [29] and AGV equation reduces to VTF Equation (12) as [58]
τ = τ_o_ exp [D·T_0_/(T − T_0_)](15)

Equation (12) is known as the modified VTF with D as the strength parameter and is used to describe relaxation above T_g_. The VTF temperature of structural arrest, T_0_ is determined as
(16)$$\frac{\Delta H_{m}}{T_{m}}=\int_{To}^{Tm}\frac{\Delta Cp}{T}dt$$
where ΔH_m_ is the enthalpy of fusion. For many fragile glass-formers the T_0_ is very close to T_K_ obtained from calorimetric measurements (typically 0.9 < T_K_/T_0_ < 1.1) [29].

## 3. Product Development Strategy

Solid dispersions have been classified in various manners based on the type of carrier (small molecules such as urea, sugars and macromolecules such as polymers) [36] or type of interaction or structure between the components (i.e., co-precipitate, solid solution, amorphous glass) [28].

The goal in ASD development is to convert the drug to an amorphous form and uniformly disperse it in the polymeric substrate in as finely a sub-divided state as possible, ideally as a molecularly dispersed state. The various stages involved in development of an ASD are outlined in Figure 7, where some of the theory described in the previous section can be applied.

At a conceptual stage, the suitability of converting crystalline drug to amorphous form or ASD is evaluated in a high-throughput screening mode by combining with different polymeric stabilizers and/or solvents under different conditions of heating and cooling followed by removal of solvent and moisture, when required. The ASD samples with different drug content (i.e., drug loading) are evaluated at a small scale (µg to mg) for phase inversion, affinity to the polymer, supersaturation limit against crystalline form and storage stability. Solid-state properties such as glass transition temperature, enthalpy of relaxation, miscibility parameter and potential for recrystallization [60] are determined using various analytical techniques including thermal, microscopy and spectroscopy. Based on these studies, the polymer(s) that can provide the most stable ASD and optimal drug–polymer ratio are identified. This is followed by selection and optimization of the process technology and parameters for larger scale preparation of the ASD as a drug intermediate as outlined in Figure 8. This step is one of the most critical due to the impact of manufacturing stress on quality, stability and performance of the ASD from the time of manufacturing up to the shelf-life of the drug product. The critical quality attributes (CQAs) of the ASD, including stability upon storage and during transportation are determined to support storage conditions and shelf-life assignment. The ASD as an intermediate is formulated into the final market dosage form such as a compressed tablet or capsule or powder filled into a sachet depending on the dosage, dosage regimen and patient-centric needs (i.e., pediatric, in-home, hospital use, etc.). These downstream steps are influenced by the physical and mechanical properties of the ASD intermediate and its quantitative content in the market formulation as described later in Section 4.

### 3.1. Formulation Development

#### 3.1.1. Drug Substance Considerations

Drug substance properties such as solubility, pKa, LogP and crystallization tendency are critical for success of ASD formation, maximum drug loading capacity and stability of ASDs. In pH-based precipitation studies of weak bases, dipyridamole exhibited a higher maximum supersaturation (ratio of peak conc. to equilibrium solubility) compared to loratadine (12.5 vs. 6.7) but the duration of supersaturation for dipyridamole was about 15 min compared to more than an hour for loratadine [61].

In a similar precipitation study of structural analogues of dihydropyridine class of drugs, the crystallization tendency of the molecules from supersaturated solutions was found to be highly variable ranging from immediate to several hours [62]. This suggests that the crystallization tendency of a drug molecule is an important aspect in ASD development. Drug substances with slower crystallization tendency exhibit greater stability of amorphous state, higher solubility and longer duration of supersaturation levels. In an ideal situation the amorphous stability of a drug in aqueous environment is extensive enough to measure its ‘amorphous solubility’ based on ‘liquid–liquid phase separation’ (LLPS) evaluation, while less ideally, polymer is required to prevent crystallization and desupersaturation during such investigations, as was shown for felodipine and nifedipine [63]. It can be supposed that the less stabilization against crystallization is required the less absorption limitations might be encountered.

#### 3.1.2. Polymeric Stabilization of ASD

##### Selection of Polymeric Stabilizers in ASD

The stabilizing polymer plays a key role in limiting the molecular mobility of the amorphous drug and stabilizing it in a high energy state in the ASD. Therefore, polymer selection is a further critical aspect of ASD formulation development. The molecular weight, T_g_, chemical reactivity, impurity profile, hygroscopicity and pH solubility of the polymer are factors of importance. For oral formulations, polymers with a GRAS status and high intake limits (e.g., >100 mg/kg body weight) are desirable. Three typical classes of polymers are routinely employed in ASD development:Cellulose derivatives (ethers/esters) are preferred due to their wide safety and functionalities obtained with alkyl-substituted hydroxyl groups e.g., methoxy, hydroxypropyl, phthaloyl, acetyl and succinyl groups that provide different aqueous dissolution profiles (e.g., hypromellose, HPMC-AS);Polyvinylpyrrolidones and vinyl acetate balanced co-polymers are also used in ASDs due to their relative ease in processing. Their functionalities are limited to balancing the hydrophilic and hydrophobic monomers and degree of polymerization (e.g., PVP, copovidone);Methacrylic acid and methacrylate esters are another class of polymers that offer functionalities tailored for dissolution at different pH of the aqueous medium e.g., with containing carboxyl or amino groups (e.g., Eudragit L or E).

Polymers utilized in ASD exhibit a wide range of glass transition temperature, T_g,_ from ca. 50 °C to 200 °C depending on the chemistry and molecular weight. A higher T_g_ suggests a relatively lower mobility of the drug at a given storage temperature and is therefore desirable. Cellulosic, non-ionic polymers such as HPMC exhibit a high T_g_ (150–180 °C) followed by anionic polymers such as HPMC-AS (120–140 °C), polyvinyl, non-ionic copovidone (ca. 100 °C), anionic, methacrylic acid copolymer such as Eudragit L (>110 °C) and cationic methacrylate copolymer such as Eudragit E (ca. 50 °C) [64,65,66,67]. PEG-vinyl based graft copolymers such as Soluplus also exhibit low T_g_ (ca. 70 °C), likely due to the PEG component.

The functional groups of the monomer unit, the molecular weight and configuration (e.g., linear vs. branched chain) determine the overall chemical reactivity of the polymer. Although a drug–polymer interaction (hydrogen bonding, ionic or hydrophobic forces) that reduces the free energy of the system and the driving force to recrystallize is desirable, chemical reactivity between drug and functional groups in the polymer is not desirable as it can lead to the generation of potentially hazardous degradants during manufacturing of the ASD under certain processing conditions (e.g., high temperature, residual moisture). Polymers with a low reactivity and low levels of reactive impurities are preferred. Vinyl polymers such as povidone and copovidone with residual peroxides can destabilize an ASD via oxidative drug degradation. In such cases, use of specific grades with low levels of peroxides or antioxidants for protection against oxidation are desirable.

The hygroscopicity of the polymer in ASD and the hydrophobicity of the drug can potentially influence its moisture sorption and stability depending on the drug–polymer ratio and exposure to moisture during storage. Polymers with low hygroscopicity are preferred in the order of Eudragit/HPMC-AS < HPMC/Soluplus < copovidone < povidone [68,69]. In the authors’ work (unpublished results), ASDs containing 30% drug and 70% HPMC (ASD 1) or 70% Soluplus (ASD 2), respectively, were stable amorphous powders at ambient conditions of storage. However, when exposed to 40 °C/75% RH for 3 months, ASD 1 remained amorphous while ASD 2 exhibited crystalline peaks. This was attributed to the low T_g_ of Soluplus and moisture sorption inducing higher mobility of the drug compared to HPMC. Interestingly, similar ASD formulations containing 50% of another similarly hydrophobic drug in the same polymer systems were stable under similar conditions of storage, suggesting that poorly water soluble drugs may stabilize ASDs through their hydrophobicity by limiting water absorption and plasticization.

Finally, the pH dependency of polymers enables tailoring drug release in specific regions of the GI tract. A pH-tailored release of the drug from the ASD at the target region of absorption can mitigate the risk of nucleation and crystallization over time. Despite nucleation of organic crystals being still a relatively poorly understood matter [70,71,72,73], polymers serve as the most universal nucleation inhibitors [74,75,76,77]. In addition to the polymer, additives such as plasticizers, surfactants, antioxidants, etc. might be included as processing aids and stabilizers.

##### Plasticization

Plasticizers are additives that improve the ductility of polymers and render them more flexible or easy to deform during processing by lowering the T_g_ of the polymer. In drug–polymer ASD containing drugs that exhibit a fairly low T_g_, the polymer increases the T_g_ of the mixture resulting in an antiplasticizing effect, the magnitude of which is often dependent on the drug–polymer ratio by weight as shown by the Gordon–Taylor equation in the following section. In processes that employ fusion or melting such as hot-melt extrusion (HME), addition of plasticizer can improve flow of the melt by reducing its viscosity. This, in turn can lead to processing at lower temperatures and reduce the risk of thermal degradation, but the impact of lower T_g_ on ASD stability should be carefully evaluated. Additionally, as water is a plasticizer, sorbed water can lower the viscosity of a solid, decreasing its T_g_ and solid-state stability. High levels of moisture sorption can lead to a T_g_ around or below storage temperature and the ASD may sinter or deliquesce.

##### Drug–Polymer Interactions in ASD

In the formation of ASD, drug (A) and polymer (B) are mixed to form a homogenous mixture via suitable means (fusion, solution) followed by quenching or evaporation to obtain the amorphous form. In the initial stages of mixing, the polymer chains reach a greater state of disorder due to their weak, secondary intermolecular bonds and complex three-dimensional conformation with numerous interchain or intrachain cross links, thus enabling greater interaction with the drug. As the temperature decreases, the system undergoes “structural arrest” due to decrease in mobility near T_g_. Polymers often facilitate this structural arrest and further reduce the mobility as the polymer chains become rigid with decreasing free volume [59]. When a drug is dispersed in a polymer matrix, several homonuclear and heteronuclear interactions between the drug and polymer come into play, the relative strength of these interactions determine the final volume of ASDs, depending on whether the total interaction is greater than, less than or equal to the sum of the two [32].

The drug–polymer affinity can be determined in-silico using solubility parameters such as Hildebrand and Hansen parameters based on group contributions from various functional moieties in the drug molecule and polymer unit. The more soluble, hydrophilic polymers tend to exhibit a higher solubility parameter that decreases as hydrophilicity decreases. The hydrophilicity/hydrophobicity ratio of polymers was found to impact both the drug–polymer affinity and rate of crystallization. Strongly hydrophilic or hydrophobic polymers were found to be ineffective in delaying drug crystallization while, polymers with intermediate hydrophilicity/hydrophobicity ratio (such as copovidone) were found to substantially delay crystallization [62]. The Hildbrand solubility parameter of PVP is 23.7 but this value decreases, as PVP is cross-linked with the less hydrophilic vinyl acetate (VA) resulting in 18.3 and 14.3 for PVP/VA 64 and PVP/VA 37, respectively, and finally reduces to 10.2 for PVA. The solubility parameters of carbamazepine and dipyridamole are 24.8 and 29.6 indicating greater affinity for highly hydrophilic polymers like PVP with a high solubility parameter. However, the ASD of the drugs was found to be more stable when formulated with PVP/VA 64 (copovidone) [78]. These studies suggest that polymers need to have affinity for both the drug-rich precipitate and the aqueous phase in order to be effective crystallization inhibitors.

The primary mechanism of drug–polymer interaction in an ASD occurs via hydrogen bonding between nucleophilic centers such as amines in the drug molecule with hydroxyl, carbonyl and carboxyl groups in cellulosic, vinyl and methacrylate polymers used as carriers. FT-IR scan of a spray dried ASD of griseofulvin with HPMCAS was used to demonstrate hydrogen bonding between a C=O group in griseofulvin and –OH groups in HPMCAS [62]. Hydrogen bonding between PVP and indomethacin was considered to inhibit formation of dimers of indomethacin [63]. The interaction between ASD of dipyridamole and PVP was determined as hydrogen bonding using FTIR where a significant effect of dipyridamole on C=O conjugation band was observed [64].

The glass transition temperature, T_g_ is one measure of drug–polymer interaction in an ASD. The Gordon-Taylor equation [65] based on the additivity of free volumes of individual components can be used to estimate the T_g_ of a binary mixture as
T_gmix_ = (w_1_ T_g1_ + Kw_2_ T_g2_)/(w_1_ + Kw_2_)(17)
where w is the weight fraction, subscripts 1 and 2 refer to drug and polymer, respectively, and K is a GT constant depending on the interaction density between the drug and polymer, calculated using the Simha–Boyer rule [66]. Deviations of T_g_ mix (experimental vs. predicted) suggest contraction or expansion of the free volume influenced by drug–polymer affinity [32] and the entropy of mixing [32,67]. A negative deviation (lower T_g_ mix than predicted) suggests a decrease in rigidity of the polymer due to plasticization effect of the drug [19].

The Ising model [68] is a simple description of two or more interacting systems with an interaction strength function, J
J = (ϵAA + ϵBB − 2ϵAB)/4(18)
where ϵ is the interaction energy between same (AA, BB) or different species (AB). The free energy change arising from such an interaction is described by the Flory–Huggins theory based on entropy of mixing of polymer dissolved in a solvent [69]. This approach is adapted for mixing of drug A in polymer B at temperature T where the free energy change in mixing, ΔG_m_ is expressed using the Flory–Huggins interaction parameter, χ, as follows:
(19)$$\Delta G_{m}/RT\;=\;(\underbrace{n_{A}ln\varphi_{A}\;+\;\varphi_{B}\cdot ln\varphi_{B}\;}+\;\underbrace{\chi \;n_{A}\;\varphi_{B})}\;\mathrm{Entropy}\;\mathrm{of}\;\mathrm{mixing}\;\mathrm{Enthalpic}\;\mathrm{component}$$
n_A_ and φ_A_ are number of moles and volume fraction of drug, n_B_ and φ_B_ are number of moles and volume fraction of polymer, respectively (φ_A_ + φ_B_ = 1) and R is the universal gas constant. 

Substituting for chemical potential of drug, μA, on a molar basis
Δμ_A_/RT = ln(1 − φ_B_) + φ_B_ + χ φ_B_^2^
(20)
χ = ΔH_m_/(k_B_ Tn_A_ φ_B_) (21)
where k_B_ is Boltzmann’s constant. The interaction parameter χ can be determined using data from melting point depression studies [70] and via solubility parameters that employ group contribution methods [60,71]. The interaction parameter is a function of both temperature and concentration of the drug in the polymer. The enthalpy, entropy and free energy of mixing curves against different concentrations of drug and polymer are used to construct the phase diagrams representing kinetics of phase separation as seen in Figure 9. The binodal boundary curves separate homogenous regions from heterogeneous ones, while spinodal boundary curve separates metastable and unstable regions.

#### 3.1.3. Early Phase Testing

In early development, the focus is on identification of physically and chemically stable ASD compositions as indicated above, i.e., evaluation of drug load and miscibility with polymers. To this aim, technical samples of ASD powders are subjected to accelerated stress testing, e.g., several weeks storage at 40 °C/75% relative humidity (RH). The samples are analyzed for chemical purity, amorphous state by pXRD, moisture content using TGA or Karl–Fischer and thermal properties such as glass transition T_g_ using modulated DSC (mDSC). Ideally, a single T_g_ should be observed. Multiple glass transition points, T_g_ could be related to phase separation or lack of miscibility. Drug dissolution studies are performed to evaluate if the drug release from the ASD is favorable, i.e., supersaturation is maintained according to the spring–parachute concept [80]. However, strong solubilization or other interactions could lead to reduced permeation/absorption. Therefore, dissolution might be combined with flux or permeability testing [81,82,83,84]. In order to simulate the variation in human gastrointestinal pH, a two-stage dissolution with, for example, initial 30 min in 0.1 N HCl or simulated gastric fluid (SGF), followed by transition into fasted state simulated intestinal fluid (FaSSIF) can be employed. The techniques of ASD characterization are detailed in the next section.

### 3.2. Analytical Testing and Characterization of ASD

#### 3.2.1. Solid-State Characterization

Amorphous solid dispersion characterization requires numerous orthogonal techniques in order to investigate drug–polymer interactions that influence various parameters such as drug–polymer affinity and miscibility, glass transition temperature of the ASD, molecular mobility and detection of potential crystallization. The purpose of performing the analysis is to demonstrate that the amorphous state of the drug substance can be achieved during manufacturing and maintained during the shelf-life of the drug product [60].

Thermal techniques are classical methods for ASD characterization with DSC and mDSC commonly used for detection of desorption, T_g_, (re)crystallization, melting and decomposition. mDSC is particularly useful for separating the overlapping thermal events and T_g_ evaluation [68]. Thermal methods with more elaborate cycling regimes can provide information on ASD formation and stability. With short run times and possibility of automation, they are ideal for initial miscibility studies as well as for stability evaluation.

The analysis of mechanical properties of ASD is desirable as they are influenced by the interactions in the ASD at the molecular level. The Dynamic Mechanical Analyzer (DMA) is commonly used for this purpose. Small cycling loads are applied on the samples enabling determination of viscous and elastic components of the complex modulus and by simultaneous temperature variation an exact T_g_ value can be obtained [68,85]. The technique can also be adapted with a chamber for relative humidity variation which enables assessment of humidity and its plasticizing effects. DMA is ideally suited for evaluation of HME extrudates [85], but also used for thin film ASD prepared by solvent evaporation. DMA can be used for identification of proper HME process parameters to obtain homogenous extrudates and to assess molecular mobility which can be linked to ASD physical stability [85].

Optical methods such as polarized light optical microscopy as well as hot-stage microscopy are powerful tools highly useful in the initial stages of miscibility study; they enable detection of undissolved or recrystallized crystals [60]. Another powerful tool is optical microscopy with micro Raman, which is capable of detecting small crystals on a very limited sample area [68]. Further developments in optical imaging such as second-order nonlinear optical imaging of chiral crystals (SONICC) provide new possibilities for ASD evaluation [86]. Optical microscopy is also utilized to observe dissolution of ASD particles in aqueous media. As shown in Figure 10, two simultaneous processes can be observed: ASD particle dissolution (reduction in size) and drug substance precipitation in form of fine particles on top of the ASD particles (thin black border on particle perimeter) and in water media (dark yellowish mist).

The most direct method for determining the amorphous state of the ASD is powder X-ray diffraction (XRD) [37,60] which can typically detect down to 1% *w*/*w* crystalline phase of investigated sample. This limit can be further increased by using synchrotron XRD where limit of detection can go as low as 0.1% or even 0.01% *w*/*w*.

Dynamic Vapor Sorption (DVS) is a technique for determining the hygroscopicity of the ASD. Polymers used in ASD typically have higher hygroscopicity compared to the drug substance and the moisture absorbed by ASD can greatly decrease the T_g_ leading to potential instabilities [60,87]. DVS data are highly valuable since results can guide the selection of manufacturing environmental conditions and also packaging configuration (e.g., use of desiccants).

Numerous spectroscopic techniques are available such as Raman and FTIR spectroscopy or solid-state NMR. Typically, they provide information on molecular interactions via bond correlation and are thus suitable for determination of dissolved state of the drug substance in polymer carrier and its phase homogeneity [60,68]. In general, any spectroscopic method capable of detecting molecular interactions can be applied in ASD evaluation. One such example is Nuclear Quadrupole Resonance (NQR) which detects the distance between N atoms and therefore is suited for evaluation of ASD of drug molecule with a large number of N atoms.

Various electron microscopic techniques can enable the detection of crystals in ASD: Scanning Electron Microscopy (SEM) images with high magnifications that project the surface texture in great detail provide a good method for observation of potential crystallization; when upgraded with the Energy-dispersive X-ray spectroscopy (EDS), the grouping of elemental composition can further aid the detection of crystals [60]. Furthermore, Transmission Electron Microscopy (TEM) with substantially higher resolution can also be used to further enhance the detectability of phase separation. One of the numerous possibilities with TEM is also the measurement of Selected Area Electron Diffraction (SAED) pattern, which enables crystallinity measurements of areas less than 1 µm in size [88]. Atomic Force Microscopy (AFM) is a resource intensive technique but can detect crystallized material in the ASD film with a resolution of up to 1 nm [60].

#### 3.2.2. Dissolution Testing and Drug Release from ASD

The purpose of ASD is to sustain in-vivo supersaturation state sufficiently long to enable absorption of poorly water soluble drugs (BCS II and BCS IV) to the extent that therapeutic drug levels are attained. The role of a dissolution test in ASD characterization is thus to demonstrate stability of the amorphous form during manufacturing, packaging and storage across shelf-life and to mimic in-vivo bio-performance of the product in order to possibly establish IVIVC. Drug release from ASD is in complex dynamic equilibrium depending on properties of undissolved, dissolved, precipitated drug, drug in drug-rich micro-domains (such as nano-droplets, micellar nanoaggregates, etc.) and on the properties of polymer, media for dissolution and interactions among different species (i.e., drug–polymer, drug–media, polymer–media interactions) [89,90]. Techniques beyond simple dissolution tests are needed in order to understand and characterize the drug release aspects of ASDs.

Since quality control (QC) dissolution tests are conducted under sink or near-sink conditions, a supersaturation state is difficult to demonstrate with a QC test and the efficacy of excipients as solubility enhancers or precipitation inhibitors or other ASD advantages cannot be assessed. Usually, QC methods discriminate between 100% amorphous sample and amorphous samples spiked with various levels of crystalline substance and between other identified CPPs/CMAs as applicable. QC methods must also show a certain level of robustness as well as inter-laboratory transferability. Due to the inherent limitations of QC methods, non-sink, tailor-made dissolution approaches along with possibility of including an absorptive interface (i.e., simulating dissolution and permeation simultaneously) integrated with discriminating analytical methods (i.e., solid state characterization) are needed as in-vitro tools to ascertain phase-behavior of ASDs in dissolution media. By careful selection of in-vitro experimental conditions such as the type (i.e., pH, ionic strength, surfactants) and volume of dissolution media, agitation speed, temperature and sampling procedures, the extent of supersaturation and formation of crystalline seeds can be captured [90,91]. Dissolution can be performed in one-, two- or multi-compartment experimental design (to simulate partitioning in the GI and systemic circulation) or in a custom-made technical design, such as TNO intestinal model 1 (TIM-1) or USP II apparatus coupled with peristaltic pumps to simulate flow of different intestinal fluids to the site of drug release and absorption [92]. The complexity of an in-vitro dissolution system therefore depends on the extent of detailed simulation of in-vivo gut physiology that is desired to be reproduced with the in-vitro set-up.

Regardless of the dissolution design, a single dissolution media or different media can be used during dissolution test to simulate shifts in pH and fluid composition in the GI tract with the possibility of including bile salts and/or surfactants. In addition, oil-like media can be included on top of an aqueous buffer to sustain quasi-sink conditions or a semi-permeable membrane (i.e., PAMPA membranes, CaCo-2 cell monolayers, microFlux system) can be included to simulate the absorption barrier [91]. A review of recent publications on analytical and dissolution techniques including those from FDA database and bio-relevant dissolution methods to assess solid dispersion formulations is presented in Table 2 below along with a summary of the composition of solid dispersion and preparation process. In addition to dissolution, other parameters such as supersaturation and solid-state properties were also monitored to assess performance of solid dispersion. While most of the articles focused on specific formulations, a few also provided dissolution data in comparison with the listed reference product. No in-vivo (animal or human) data were presented in any of the publications included in Table 2.

To demonstrate practical applicability of dissolution in ASD development, herein is presented a case study of ASD formulation designed as a prolonged-release product. A generic ASD was prepared by fluid-bed granulation with subsequent milling to granular size that provided the targeted drug release. The drug is poorly soluble in aqueous media (BCS II) and patients receiving it necessitate therapeutic drug monitoring. A reliable dissolution method was required for such a product in order to support timely manufacture of the product and to assure patient safety and efficacy. Dissolution methods that provided sink conditions (i.e., using surfactants) were not applicable in this case because they lacked sufficient discriminatory power towards identified CPPs and CMAs. Therefore, a dissolution method using aqueous solution of polyethylene glycol as a medium was developed and found to be discriminatory towards qualitative/quantitative composition, granule particle size and temperature of the product during granulation among others. To establish in-vitro release, two samples of similar granule particle size (d_10_, d_50_ and d_90_ at 100, 180 and 270 µm for Sample 1 and 113, 190 and 290 µm for Sample 2, respectively) prepared at different granulation temperatures were tested in-vivo for similarity. Based on in-vitro release profiles shown in Figure 11 and in-vivo results (C_max_ and AUC for Sample 1 at 101% and 115% of reference and for Sample 2 at 99% and 102% of reference, respectively), IVIVC was successfully established.

## 4. Manufacturing Technologies and Scale-Up

Commercial scale manufacturing of large quantities of ASD up to several hundred kilograms requires a technology with optimized process and operational parameters that provides the maximum product yield meeting the required critical quality and safety attributes (CQAs) through its shelf-life. During scale-up of the product, QbD tools such as DoE and PAT are often employed to identify optimal design space of critical material attributes (CMAs) and critical process parameters (CPPs) to obtain a product with the desired CQAs. The technology and associated processes must be robust, reproducible, scalable and amenable to operations and qualifications under cGMP. The production of ASD is therefore limited to a few technologies meeting these requirements for large-scale operations and control strategies.

The impact of the manufacturing process on ASD quality and stability is a key consideration in scale-up. While the stability of amorphous form in the ASD is increased with reduced molecular mobility, certain factors such as high temperature during processing, high humidity and residual solvents enable plasticization of ASD, increasing its mobility and risk of crystallization. Being more reactive than its stable crystalline form, the amorphous form in an ASD is more susceptible to chemical degradation via mechanisms of oxidation and hydrolysis linked to manufacturing processes such as solvent evaporation and melting leading to a lower potency of the drug in the dosage form, increase in the impurity content and/or the risk of crystallization by plasticization due to lowering of the T_g_.

The risks related to residual moisture content and solvents are also applicable to the downstream processing of the ASD into a dosage form suitable for market. Dry blending and granulation processes such as roller compaction are preferred over wet granulation in order to increase the bulk density and the flow properties of the ASD intermediate [108]. While the use of liquids such as water or organic solvents in the wet granulation poses a similar comparable risk as that of solvent evaporation methods [109,110], examples of wet granulation and similar aggregation technologies to improve flow and compressibility of solid dispersion formulations have been reported [111].

In addition to the aspects related to the manufacturing processes, environmental factors such as room temperature and humidity during processing need to be controlled in order to maintain the stability of ASD and minimize the impact on the supply chain of the product. Approaches to mitigate risks related to environmental factors during processing and storage of ASD include:Maintaining room humidity conditions at a relatively low level <40% RH;Packaging that prevents or significantly reduces moisture penetration such as sealed Alu bags for bulk storage, glass/HDPE bottles with desiccant and Alu-Alu blister packaging;Storage and transport under “cold/refrigerated” conditions to maintain ASD stability (cold storage reduces mobility via lower entropy) especially for handling in tropical countries (Africa and South East Asia) classified under climactic zones III and IV.

### 4.1. Overview of ASD Manufacturing Technologies

In the early stages of drug development and formulation assessment, small amounts of ASD are prepared in the laboratory using techniques such as solvent evaporation in a rotary evaporator, film casting in a Petri dish and melt-quenching in a DSC pan [112]. As development progresses, technologies suitable for cGMP processing at the pilot and commercial scale are evaluated for manufacture of ASD. These are classified as methods based on melting and use of solvents [36,113,114] as described in Table 3.

### 4.2. Technologies Based on Melting

The fusion/melting technology was first used in 1961 by Sekiguchi and Obi [116]. It involves heating a physical mixture of a drug and hydrophilic carrier until they melt at a temperature slightly above their eutectic point. The melt is rapidly cooled and solidified in an ice bath with stirring. The resulting solid mass obtained is crushed and sieved into a powder. This method produces a soft, tacky material poorly suited for tablet compression. The fusion method is amenable for drug–polymer mixtures that form a solid dispersion as a homogenous single phase in a specific ratio at the eutectic temperature. The solid dispersions from the early 1960s and 1970s were simple eutectic or monotectic mixtures of drug and carrier such as that of sulfathiazole-urea and griseofulvin-PEG 2000–8000 [28,116,117,118] prepared by fusion or melt-congealing techniques. Another technique involves dissolving the drug and a polymer in a solvent followed by evaporating the solvent to obtain the solid dispersion as a co-precipitate. Solid dispersions of several poorly soluble drugs such as griseofulvin, sulfathiazole, reserpine, nifedipine, indomethacin and ketoprofen with PVP have been studied in this manner [119,120,121,122,123]. The product GRIS-PEG™ describes the drug as “ultra-micro size” due to the finely divided particle size of the solid dispersion that provides faster dissolution rate. HHME technology is the modern version of the fusion method in which components such as drug, polymer and plasticizer are intensely mixed under heating (melting) to obtain a homogenous mixture which is extruded through a die plate. Compared to the traditional fusion method, HME offers the potential to shape the molten drug–polymer mixture into rods, implants, pellets or milled extrudate formulated as solid dosage forms [124]. Melting methods offer the advantage of preparing ASD in a solid-state without the use of solvents or vehicles.

KinetiSol^®^ is a relatively new ASD manufacturing process that was developed to formulate high melting APIs with a polymeric carrier to form a stable ASD. Heating is not introduced externally into the process but is generated solely through friction and shearing of the materials and blades that rotate at high speeds (>1000 rpm), producing a stable ASD. The ASD is generated in shorter processing times and lower processing temperatures, offering a key advantage over conventional HME. Recent advancements in the addition of thermally conductive excipients have further increased the applicability of the KinetiSol^®^ process [125].

### 4.3. Technologies Based on Use of Solvents

The solvent evaporation method involves dissolving drug and carrier in a common solvent to form a homogeneous solution followed by evaporating the solvent under constant agitation of the solution leading to the ASD that is then crushed and sieved to obtain a solid powder. Unlike the melting method that requires the drug to be heated as a mixture at high temperature, this method is suitable for thermal-sensitive and thermolabile materials, i.e., drugs and carriers not amenable for melting method since the temperature of evaporation is usually low [126]. However, two key limitations exist with this approach. It is difficult to find a common solvent or mixture of solvents that can dissolve both drug and carrier especially if they have significant differences in polarity. Surfactants may be used to improve the drug or polymer solubility in particular solvents, but at amounts as high as 10% *w*/*w* in the final dosage form surfactants can reduce the drug loading capacity and pose safety concerns upon ingestion. The second limitation is the risk of phase-separation induced by the residual solvent if not completely removed during drying.

Different techniques are used for removal of the solvent such as slow application of heat (for small amounts of solvents in a rotary evaporator), adiabatic expansion (flash evaporation via spray drying or fluidization), sublimation under low vapor pressure (lyophilization [127]), application of electrical field on a drug–polymer solution (electrostatic spinning that generates hardened drug–polymer nanofibers [128]) or extraction of particles using supercritical fluid (use of supercritical CO_2_ for particle extraction at a temperature above its critical temperature of Tc = 304.1 K [129,130]). Additional information on the manufacturing technologies of ASD is available in the literature [130,131].

### 4.4. Current Technologies for Commercial Manufacturing of ASDs

Currently, spray drying (SD) and fluid-bed layering (FBL) technologies based on the principle of solvent evaporation and HME technology based on fusion/melting are scalable processes commonly employed in the pharmaceutical industry for commercial-scale manufacturing of ASD, with SD and HME accounting for more than 60% of marketed products as seen in Figure 12 while, Figure 13 shows the various stabilization polymers used in Marketed ASD products. While the process parameters in these technologies can be optimized using QbD tools to identify CPPs, CMAs and CQAs [124,132,133,134], the choice of technology is determined by criteria such as T_m_ of the drug substance, its solubility in the selected polymeric carrier and/or solvents and its sensitivity to thermal and/or mechanical stress based on its degradation potential [135].

#### 4.4.1. Spray Drying (SD) and Fluid-Bed Layering (FBL)

SD is a commonly used solvent evaporation method for preparation of ASD of thermolabile and high melting temperature drugs. The drug–polymer is dissolved or suspended in a common solvent or solvent mixture followed by drying over a stream of heated gas (air) introduced at a controlled rate of flow to remove the solvent. Evaporation of solvent occurs at a very fast rate in SD causing a sudden rise in viscosity which leads to the entrapment of drug molecules within the polymer matrix. For poorly water-soluble drugs, organic solvent or solvent mixture(s) is used to dissolve the drug and polymer. As with any solvent-based process, potential toxicity concerns due to handling of large amounts of solvents at commercial scale is a concern. Solvent selection is generally governed by its boiling point, solubility of drug and polymer in the solvent and toxicity of solvent as defined by ICH classification (for example, class III solvents are preferred over class I solvents due to lower toxicity potential) [136].

Several SD process parameters (CPPs) must be optimized in order to obtain a high yield of ASD with the desired CQAs of particle size, morphology, crystalline limit of the ASD and residual solvent content. The process parameters include properties of inlet air (or carrier gas) such as temperature, humidity and flow rate, attributes of feed material containing drug substance and solvent such as viscosity, surface tension, type of solvents and solids content and conditions related to spray such as spray rate of the solution, type of atomization nozzle and spray pressure [137,138].

FBL is a solvent evaporation method based on a similar principle as that of SD, i.e., adiabatic expansion with similar CPPs and CMAs as that of spray drying. A drug-carrier solution in a solvent is sprayed onto the surface of non-pareil starch or sugar spheres in a fluidized bed equipment to produce ASD-layered granules or pellets with acceptable flowability and compressibility which is suitable for tableting, coating or encapsulation [139]. Figure 14 shows microXCT photographs of SD and FLB particles with a porosity of 22.45% and 21.37%, respectively, indicating the similarity of the two methods.

#### 4.4.2. Hot-Melt Extrusion (HME)

HME offers the potential for continuous manufacturing of ASD with high yield, adequate control of operating parameters and scalability to commercial level. Furthermore, oxygen and moisture may be excluded almost completely for substances prone to oxidation and hydrolysis. Several CMAs and CQAs are identified in HME including melting temperature, T_m_, glass transition temperature, T_g_, of the carrier, drug–carrier miscibility parameter, thermal stability, drug content in ASD, melt viscosity, particle size and performance of finished dosage form (dissolution, potency, uniformity of content). Several HME process parameters (CPPs) need to be considered for scale-up of manufacturing including design of the screw elements and mode of assembly, melt temperature, screw speed, feed rate, ability to vent gases (degassing ports), configuration of die plate, moisture content and particle size of the feed blend and milling parameters [124,140,141]. In addition, environmental conditions such as relative humidity in the operational space and cooling rate are factors to consider in HME processing, especially for hygroscopic drug substances [142]. The Specific Mechanical Energy (SME) is typically a calculated parameter that represents energy expended in the HME process and is used for scale-up.

The screw elements comprise one of the most important parts of the HME equipment that can impact the quality of the final product. The design and placement of the screw elements in the appropriate configuration and the proper ratio between conveying and kneading elements represent a key aspect in set-up of the extrusion process in HME. Kaletra^®^, developed by Soliqs (Meltrex^TM^) became the first marketed ASD using HME that led to the twin-screw co-rotating extruder being one of the most widely used HME technologies for commercial manufacturing of ASD. Compared to the single-screw configuration, the twin-screw mode reduces the material residence time and thermal stress on the material being extruded. In addition, the “self-wiping” mechanism of the co-rotating twin screws ensures self-cleaning of the screws during the process. In a co-rotating and intermeshing twin-screw extruder, one crest edge of the screw wipes the flanks of the other screw, ensuring a more efficient and uniform self-cleaning, thus minimizing the sticking of the product on the screw surface as shown in Figure 15 [143]. Milling of the extruded material post-extrusion could be challenging due to its high mechanical resistance. In such cases, cryogenic milling may be considered [144].

#### 4.4.3. Impact of Physico-Mechanical Properties of ASD on Manufacturing Processes

The physical and mechanical properties of an ASD play a significant role in its ability to withstand downstream operations such as milling, blending and compression during the development of pharmaceutical dosage forms such as a tablet or capsule. Several factors including properties of the drug and polymer in the ASD, drug loading, storage conditions and technologies to manufacture ASD play a role in determining its physico-mechanical properties. These properties influence powder flow and compression and its deformation behavior under an applied stress. The impact of such properties on material behavior during galenical processing has been demonstrated by instrumented tablet press, compaction simulator and mechanical testing devices such as indentation hardness testers and diametral compression testing [145,146,147].

Most polymers used in ASDs consist of crystalline and amorphous regions with increasing crystalline region providing greater hardness and tensile strength while the amorphous region imparts flexibility and fluidity [148]. The melting, shearing and evaporation processes involved in preparing ASDs can result in plasticization of the large molecule polymer by the small molecule drug in the ASD, lowering the T_g_ and melt viscosity by increasing the free volume between the polymer chains [149,150]. These effects, in turn can alter the polymer’s mechanical properties such as tensile strength and moduli that affect the processing of finished dosage form.

The flow and compressibility properties of ASD particles obtained from SD and HME influence the downstream processes employed in conversion of ASD to a finished dosage form. Significant differences in size and morphology have been observed between particles obtained from SD and HME [151]. Most studies comparing physical properties of ASDs prepared by SD and HME [152,153,154] seem to confirm the poor flowability and higher cohesiveness of the SD powders compared with those by milled extrudate from HME. SD provides smaller, more porous and spherical particles with low bulk density that provide a faster drug release due to a large surface area, but result in poor flowability of the spray dried powder. The poor flowability and highly cohesive nature of spray dried particles were confirmed using both rheometric parameters (i.e., stability index, flow function, aeration ratio, flow rate index) obtained from FT4 analysis [155,156] and common indices such as Carr Index and Hausner ratio with comparable results. A large surface area can impact stability of the amorphous drug due to lowering of T_g_ at the surface of ASD particles compared to the bulk region [144]. ASD particles obtained from SD are often densified via granulation to improve flowability and bulk density for unit dose (tablet or capsule) development. ASD tablets prepared from spray dried particles exhibit greater tensile strength, likely due to the larger bonding area from the smaller particles [157]. However, the impact of ASD granulation on stability must be considered. In a similar approach, SD and co-precipitation techniques were used in a comparative approach to prepare ASD containing 50% of a development compound with HPMC-AS. The co-precipitation by using resonant acoustic mixing demonstrated superior tabletability and flow properties when compared to the SD powder [158].

On the contrary, HME extrudate provides granules with a higher bulk density, superior flowability and ability to withstand mechanical stress (very low friability) upon milling. Such granules are suitable for filling in capsule or sachet/stickpack with acceptable sealing of the unit due to the very low amount of fines in the ASD particles. However, the milled extrudate granules often exhibit poor compressibility and lower tablet tensile strength requiring the use of compression aids and fillers to impart plasticity for tablet compression.

The impact of HME and SD parameters on mechanical properties of commonly used ASD polymers HPMC-AS, copovidone and their formulated blends has been studied using static and dynamic hardness parameters based on Hiestand dimensionless tableting indices or DTI [154] and tabletability curves [159]. Upon HME or SD, both polymers exhibited significantly lower tablet tensile strength (tabletability) and bonding index but HPMC-AS exhibited a higher dynamic hardness after HME, suggesting the loss of plasticity which is a desirable attribute for tablet compression [154,159]. A greater impact of HME and SD processes on mechanical properties was observed for HPMC-AS than for copovidone. In addition, tablets prepared with melt extruded mixtures of HPMC-AS and copovidone mixtures exhibited decreasing tensile strength with increasing content of HPMC-AS in the mixture as shown in Figure 16. The addition of a small molecular weight drug to the HPMC-AS-copovidone mixture further altered these properties.

The yield strength of glassy materials increases with aging time [160] suggesting an increase in hardness due to changes in material properties of ASDs as a function of storage time via structural relaxation. This phenomenon seems similar to that of work hardening that involves loss of compactibility of a material upon roller compaction [161]. In a study evaluating the mechanical properties of amorphous copovidone, it was observed that higher structural relaxation of copovidone led to increased indentation hardness, resulting in tablets with significantly lower tensile strength [162].

An appropriate selection of excipients is therefore key to successful conversion of ASD from both SD and HME into a solid dosage form for market considerations [144,154,155]. In addition to excipients, drug content in the ASD also plays a role in defining the mechanical properties of an ASD and final market dosage form. At a drug content of less than 30% *w*/*w*, acetaminophen (APAP) was found to plasticize copovidone during formation of ASD, resulting in lower T_g_ and reduced tensile strength of ASD tablet across a solid fraction of 70–90%. A decrease in T_g_ is an indication of plasticization of copovidone by APAP. This suggests greater interaction (stronger glass) between APAP and copovidone at lower drug loading resulting in a tablet with higher tensile strength [163]. An ASD with <30% drug content and 200 mg dose may result in a tablet mass of approximately 1 g which may pose difficulty in swallowing for the patient. In such a case, filling the ASD powder in a pouch or sachet may be more suitable.

### 4.5. Considerations in Scale-Up and Technology Transfer of SD

Pharmaceutical spray dryers are available in a wide range of scales: from lab units where milligrams of material can be produced to large commercial units capable of handling multiple tons of powder per day. During scale-up, some quality attributes of the product can change and there is need to understand whether these changes are acceptable. A poor scale-up strategy may lead to considerable losses of expensive materials and ultimately affect the timelines of a clinical program [164]. Before embarking on the scale-up of any process, it is highly recommended to attain a stable and robust process at lab scale in order to understand how key process parameters should be set at the larger-scale unit. On the other hand, powder properties could also be improved during scale-up. This results from the ability to produce and dry larger droplets in the larger drying chambers of the commercial units, influencing the compressibility and flowability of the material for downstream processes.

The principal goal during scale-up is to maintain product quality (to match physical and chemical attributes of particles across different scales) and maximize yield. Despite its criticality, the scale-up of SD process is still mostly empirical and based on costly experimentation and its statistical interpretation. To minimize the empiricism of such an approach, recent efforts have focused on applying mechanistic models and simulation tools to describe the process of SD. This rational approach has gained wide recognition and pharmaceutical scientists are now making use of it during development, scale-up and manufacturing [165].

In a review article for scale of the SD process [166], the authors emphasized the fundamental physical transformation of the material during SD based on atomization, drying and gas-solid interactions. They also focus on the use of mechanistic models and scale-up rules for establishing design spaces for the process variables involved in each SD transformation along with presenting a roadmap for process development and scale-up based on a rational understanding of the transformations. The roadmap for SD process scale-up consists of three stages:Identify key response variables for each transformation based on atomization, drying and gas-solid separation;Use correlations with non-dimensional parameters;Develop fundamental dimensional variables to ensure similarity across scale.

A practical and methodical scale-up of the SD process proposed by the equipment manufacturers and employed by the industry [164] is summarized in Table 4. The target is to keep the critical process parameters (CPPs) constant and adjust the ones that have the least impact on the process. In a spray dryer, the critical process parameters in order of importance are the outlet air temperature, the droplet size and the outlet vapor concentration:The *T_out_* is measured directly and is kept constant during the scale-up;The next part in scaling-up is to obtain the same particle size of powder through the scales. The particle size of dried powder is dependent on the droplet particle size. The droplet size obtained from an atomization process is estimated from empirical correlations that are often product and nozzle dependent and may not describe adequately the complex fluid dynamic processes [167]. Extrapolation should therefore be done carefully since the particle size may be valid only within the local experimental space. Nevertheless, the mean droplet diameter obtained empirically is still a good approximation and can be a good guide for scale-up purposes;The P_feed_ is the major driver in determining the particle size of the final powder. A higher relative velocity between the air and the liquid increases the shear forces and provides a higher atomizing energy, reducing the mean droplet size. The droplet (and particle) size is smaller at laboratory scale due to the relatively small drying chamber and shorter drying times. A larger droplet size requires a larger drying chamber with longer drying times to obtain larger solid particles. Naturally, this requires a reduction of the P_feed_ and typically also needs an adjustment of the operating temperatures to maintain the same residual moisture content in the final powder;The next step is to calculate the evaporation rate required at the pilot scale to achieve the same vapor concentration in the outlet air as in the laboratory scale, thus providing the same level of residual solvent in the spray dried powder at both scales. This is achieved by using a simple mass balance with the following parameters: F_feed_, total solids content in the feed and the residual water content in powder. From the evaporation rate at laboratory scale, F_feed_ and T_in_ are calculated for pilot scale. This is particularly important when producing amorphous materials since the residual solvent level in the ASD significantly affects its glass transition temperature (T_g_).

## 5. Novel ASD Polymer Considerations and Challenge

This section is designed to help scientists developing or incorporating a new or novel ASD polymers into products by providing an understand of the barriers that may inhibit their progress. Additionally, what is also described in this section is a “challenge to the scientific community” to encourage the development of novel polymers that offer differentiation by addressing key limitations with existing polymers.

The authors define a “new” polymer as an excipient that is already included in the FDA “inactive ingredient data base for approved drug products”, that is in an existing product at or below the daily quantity reported and via the same route of administration [168]. New excipients have lower regulatory burden. A “novel” excipient is defined by the authors as an inactive ingredient, or chemically modified inactive ingredient or “Generally recognized as safe” (GRAS) excipient that has not been previously used in a pharmaceutical product via the route of administration or is at higher daily quantity reported in the FDA or other regulatory excipient databases. Novel excipients have higher regulatory burden that will be discussed further in this section.

Figure 13 provides an overview of the polymers that are used in successful ASD marketed products. The table highlights that there are many successful polymers that are available at low cost, high volume, are suitable for use in established large-scale manufacturing processes and provide consistent performance (i.e., offer safety and improve product delivery). All formulation scientists and engineers will select from this established list of polymers to reduce risks and resources, unless there is a clear differentiation and need for a new or novel polymer. Differentiation should be considered as the first barrier.

The second barrier consists of the regulatory reviewing process for novel excipients which needs to be addressed for any new polymer irrespective of being classified as GRAS [169]. It is vital to recognize that GRAS status is given to the context of use of an excipient, not the ingredient itself. The regulatory authorities recognize the need for novel excipients as indispensable in development of new medicines that require advanced drug delivery systems [170,171]. However, they also have a responsibility to ensure that any novel excipient is safe and can provide consistent performance. Pottel et al. [172] recently published an article showing excipients that are considered “inactive ingredients”, that were found to have activity in therapeutic targets and against clinical safety target in-vitro assays. The authors concluded that “although most excipients deserve their status as inert, many approved excipients may directly modulate physiologically relevant targets.” This publication highlights the need to verify the safety of any new excipient.

Most major regulatory authorities have systems that allow an excipient manufacturer to submit the necessary details on their product to the regulatory authorities without disclosing proprietary information to competitors. Europe is the exception to this rule where currently a novel excipient can only be introduced as part of the drug product marketing authorization. This means the sponsor of a molecule using the excipient in their formulation is required to submit the application [170]. FDA Center for Drug Evaluation and Research is exploring a pilot program for the toxicological and quality evaluation of novel excipients so that it is not necessary for the novel excipient to be included in a finished drug product described in an IND, NDA or a BLA [171].

A final barrier consists of the qualification of any new excipient and the supplier within a pharmaceutical company quality system. This generally involves the quality organization establishing the specifications for a new excipient to verify the safety and performance of any new excipient for cGMP manufacturing. The process can take up to one year and requires substantial resources from both the supplier and pharmaceutical company. Consequently, no company will undertake the qualification of a new excipient unless it is essential for the progression of a molecule.

Despite the above barriers the authors have been involved in the evaluation of over ten novel ASD polymers. The focus was on enabling the delivery of new emerging classes of small molecules (i.e., PROTAC or bRo5) [173] which have low solubility and bioavailability challenges. These molecules are generally at the discovery optimization phase, where the barriers described above are minimal and there is a need for the highest drug loading to enable toxicology studies.

Table 5 provides a summary of the attributes (needs) used to establish if a new ASD polymer differentiate from the existing marketed polymers. To differentiate two or more of the attributes of the novel polymer must significantly exceed that of a currently available commercial ASD polymer.

It would be of great benefit to the pharmaceutical scientists, and patient community that they serve, if novel polymers can be developed that can exceed the performance of current ASD polymers for pharmaceutical development. The industry and excipient companies may want to consider this as a challenge using the model compounds suggested in Table 6 for evaluation of the novel polymers. These compounds are readily available, have clearly defined challenges and have available published in-vivo preclinical and clinical data for comparison. The compounds also represent a diverse set of chemistry space enabling a scientist to establish their novel polymer performance for different types of molecules.

## 6. Conclusions

ASD has been successfully deployed across the pharmaceutical industry to develop products that address drug solubility challenges. This platform accounts for approximately 30% of the marketed products requiring solubility enhancement and was the most frequently used technology from 2000 to 2020 (Figure 2, Source PharmaCircle™). The reasons for the successful application of ASD include: the significant theoretical understanding that can be directly applied to the design of the product and optimization of the manufacturing process, availability of different polymer stabilizers, use with commercially viable manufacturing technologies and alternative product presentations (i.e., tablet, capsule, sachet, granules, etc.) that when combined offer a versatile scalable platform that is suitable for different molecules.

The studies and observations highlighted herein demonstrate that while ASD formulation can improve solubility and bioavailability of a poorly soluble drug, a careful choice of the polymers, drug loading and manufacturing processes is needed to ensure that the resulting ASD is suitable for downstream processes utilized in dosage form development. The target dose and dosing regimen (i.e., qd, bid) as well as patient compliance factors such as convenience of dosing, constitution procedures, swallowability and pill burden are additional important factors to consider in the selection of the most suitable market dosage form for commercial use.

It is in the interest of the scientific community and the broader pharmaceutical industry to encourage the development of novel stabilization polymers that differentiate by offering greater drug loading (>50% *w*/*w*) while maintaining both chemical and physical stability during storage through the shelf-life of the product. Any novel polymer will also have to be compatible with existing commercial manufacturing technologies, non-toxic and provide equivalent or better oral bioavailability.

## Figures and Tables

**Figure 1 pharmaceutics-13-01682-f001:**
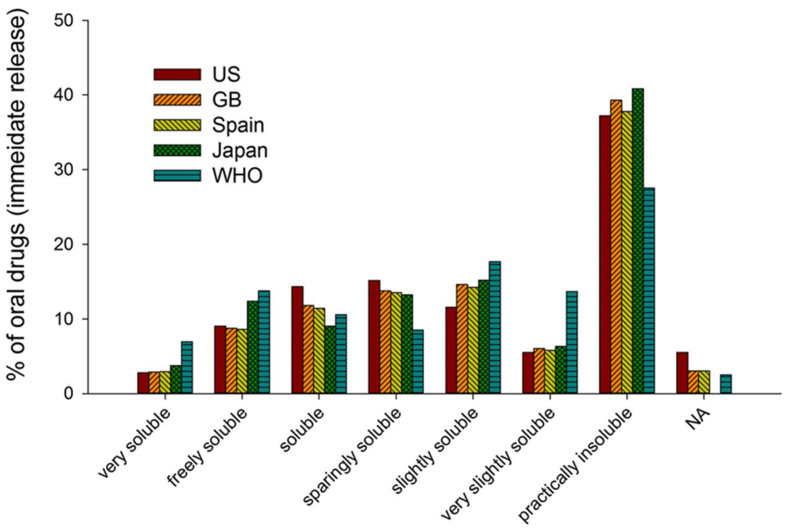
A comparison of the distribution of solubilities for 200 oral drugs from various regions of the world (very soluble drugs: over 1000 mg/mL; freely soluble drugs: 100–1000 mg/mL; soluble drugs: 33–100 mg/mL; sparingly soluble drugs: 10–33 mg/mL; slightly soluble drugs: 1–10 mg/mL; very slightly soluble drugs: 0.1–1 mg/mL; practically insoluble drugs: <0.1 mg/mL). Reproduced with permission from [4] T. Takagi et al, Molecular Pharmaceutics, published by American Chemical Society, 2006.

**Figure 2 pharmaceutics-13-01682-f002:**
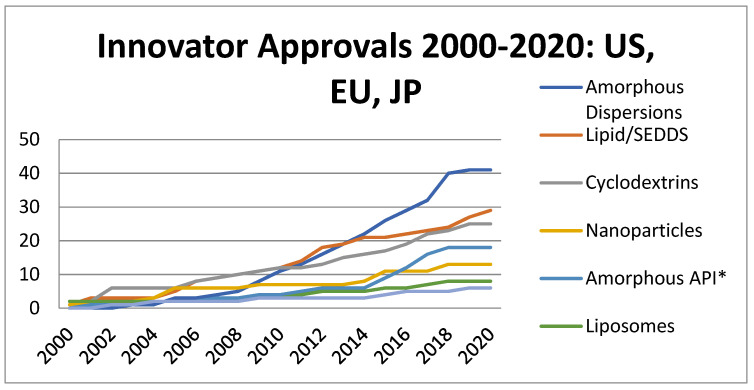
Enabling solubilization technologies used on approved marketed products from 2000 to 2020. * Drug substance formulated as amorphous form instead of ASD (source PharmaCircle™).

**Figure 3 pharmaceutics-13-01682-f003:**
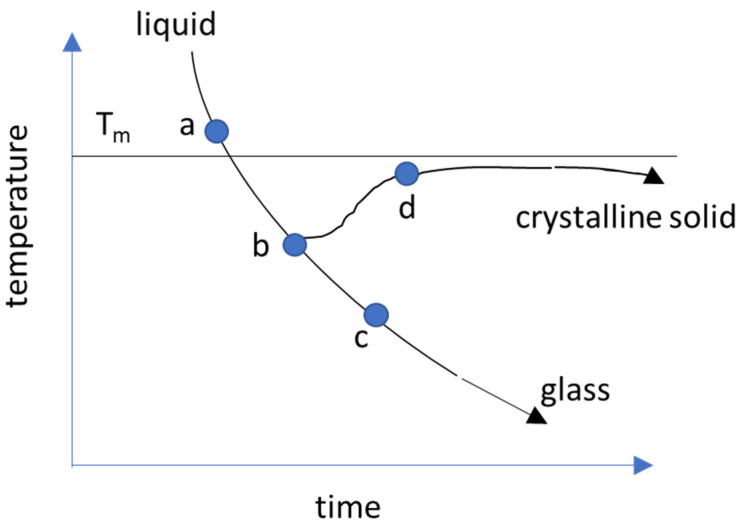
Quenching or cooling curve in glass vs. crystal formation. (Tm is the equilibrium melting temperature; a–b–d denotes path for nucleation and crystal growth; a–b–c denotes path of glass formation (Adapted with permission from [31]).

**Figure 4 pharmaceutics-13-01682-f004:**
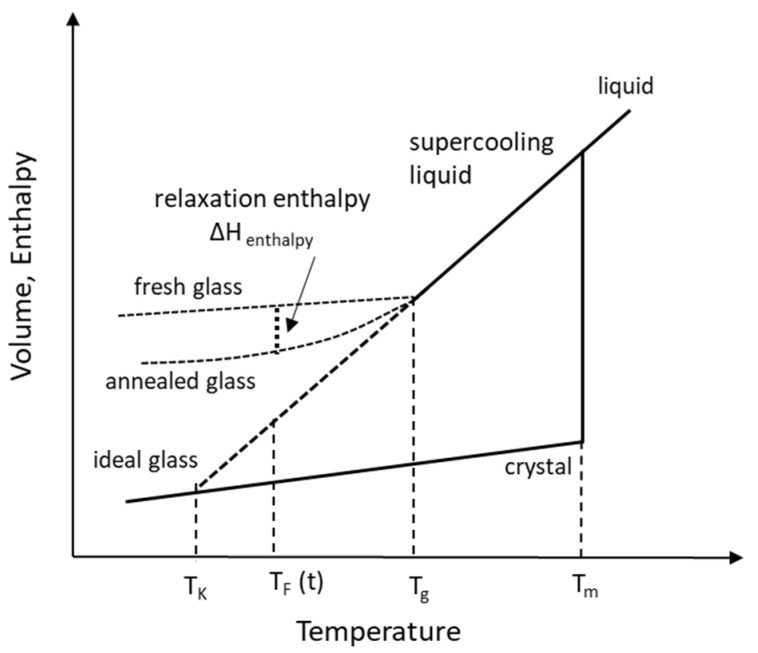
Phase diagram of liquid-crystal and liquid-glass. (Temperatures—T_K_—Kauzmann, T_g_—glass transition, T_F_—fictive, T_m_—melting).

**Figure 5 pharmaceutics-13-01682-f005:**
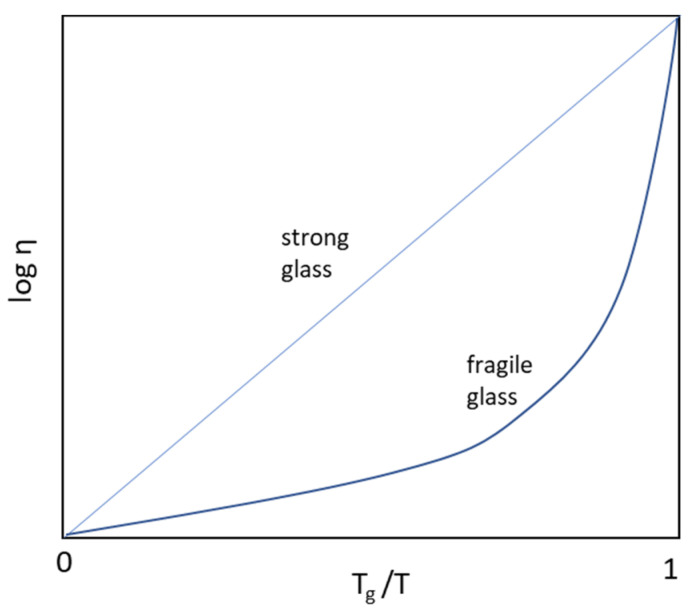
A schematic description of strong (Arrhenius type) and fragile (non-Arrhenius type) glass formers.

**Figure 6 pharmaceutics-13-01682-f006:**
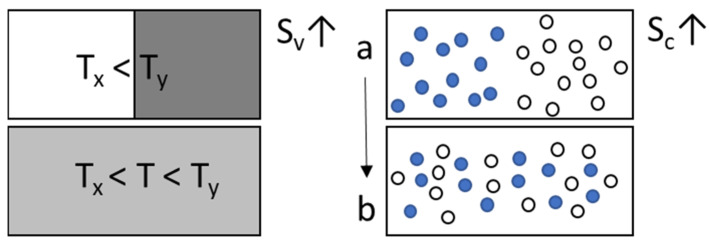
Increase in vibrational entropy between two systems at temperatures T_x_ and T_y_ as a function of temperature [S_v_ = *f* (T)] and configurational entropy as a function of distribution states from unmixed (a) to mixed (b) state [S_c_ = *f* (ln W)] (Adapted with permission from [57] Gaskell et al., Introduction to the Thermodynamics of Materials, published by Taylor & Francis Group, 2017).

**Figure 7 pharmaceutics-13-01682-f007:**
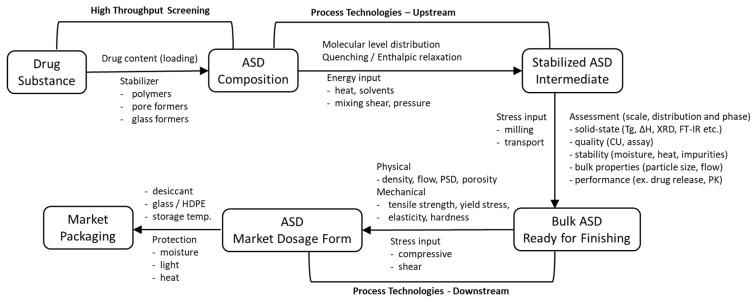
Stages in pharmaceutical development of ASD-based market and clinical formulation.

**Figure 8 pharmaceutics-13-01682-f008:**
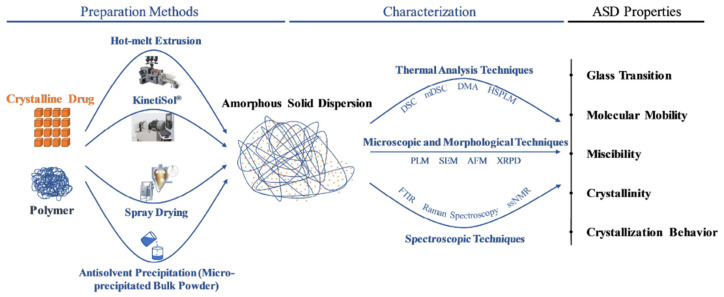
ASD preparation and characterization flow chart. Adapted with permission from [60] Ma et al., Journal of Drug Delivery Science and Technology, published by Elsevier, 2019.

**Figure 9 pharmaceutics-13-01682-f009:**
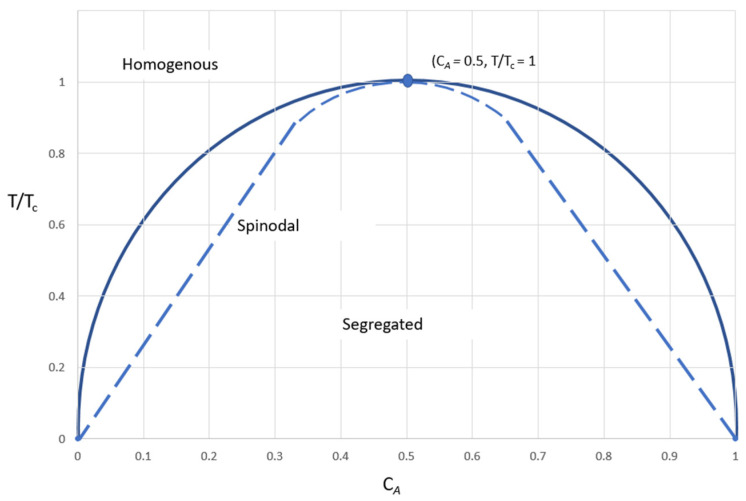
Phase diagram of a binary mixture, AB. Adapted with permission from [79] Patterson et al., Drug Development and Industrial Pharmacy, published by Taylor & Francis, 2008.

**Figure 10 pharmaceutics-13-01682-f010:**

Optical images (40× magnification) of ASD particles (produced by HME) immersed into water media at room temperature: after 1 min (**left**), 30 min (**middle**) and 60 min (**right**).

**Figure 11 pharmaceutics-13-01682-f011:**
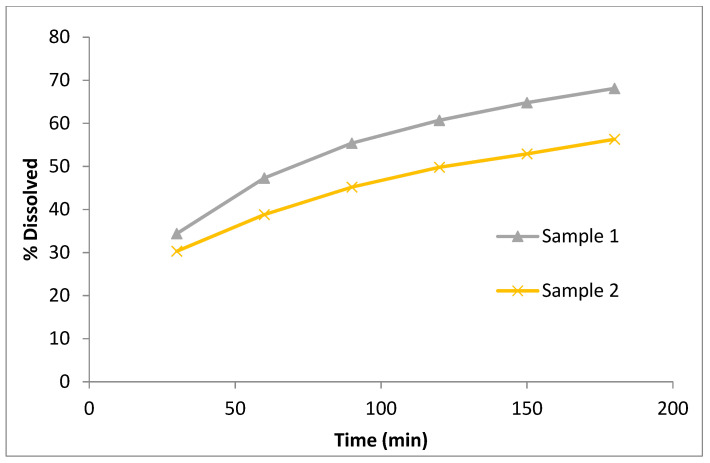
Non sink dissolution profiles for ASD of samples in fasted media.

**Figure 12 pharmaceutics-13-01682-f012:**
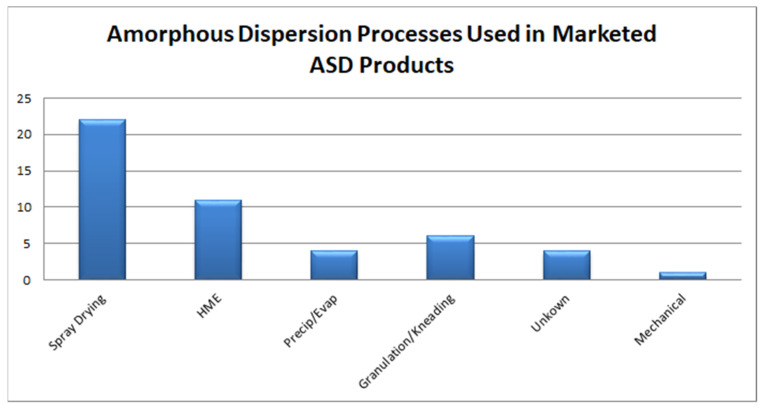
Number of marketed ASD products using different processes (source PharmaCircle™).

**Figure 13 pharmaceutics-13-01682-f013:**
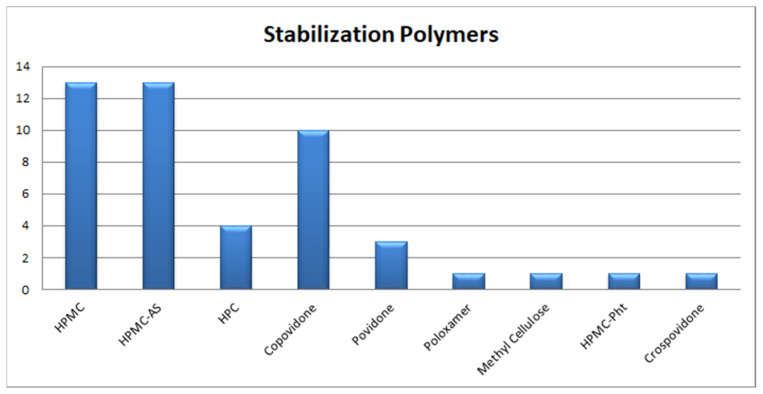
Number of stability polymers used in marketed ASD products (source PharmaCircle™).

**Figure 14 pharmaceutics-13-01682-f014:**
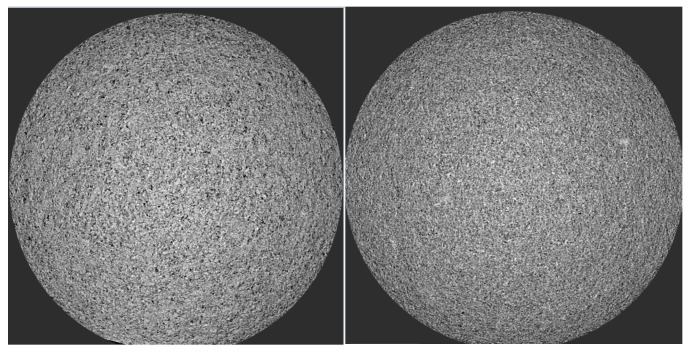
MicroXCT (4× magnification) of the spray dried material (**right**) and fluid bed material (**left**) concerning the same model drug (BCS II API).

**Figure 15 pharmaceutics-13-01682-f015:**
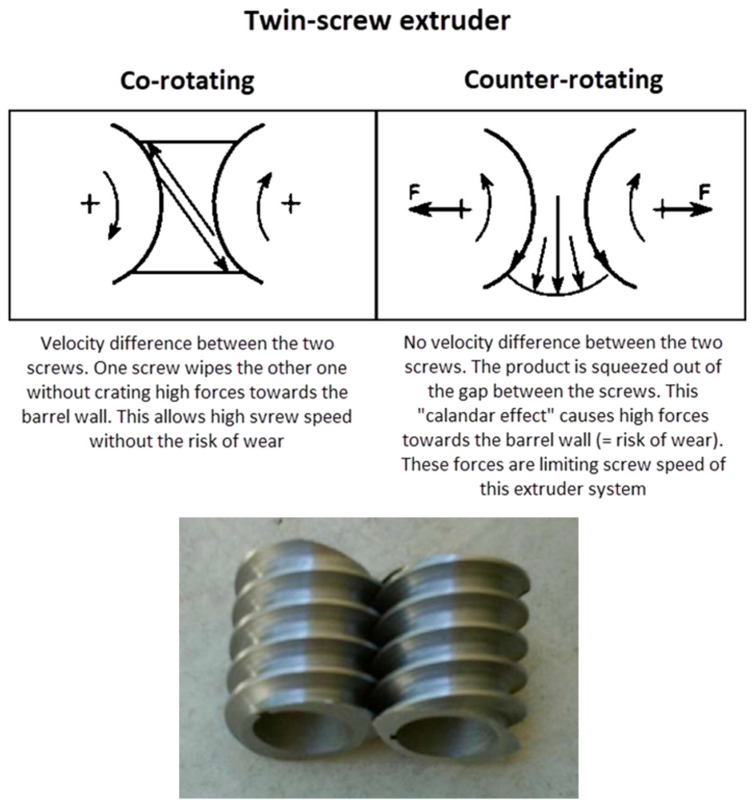
Wiping action in twin-screw extruders. Adapted with permission from [143].

**Figure 16 pharmaceutics-13-01682-f016:**
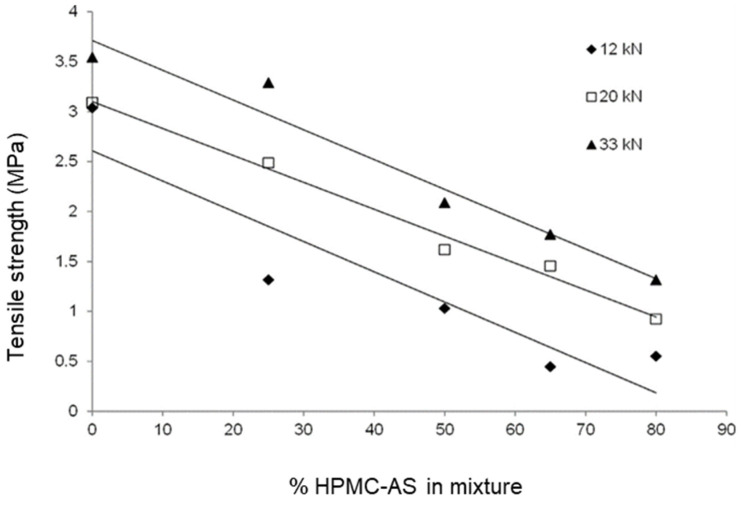
Tensile strength of ASD tablets composed of mixtures of milled extrudate of HPMC-AS and copovidone at compression forces of 12, 20 and 33 kN. Adapted with permission from [154] Iyer et al., Journal of Pharmaceutical Sciences, published by, Elsevier, 2013.

**Table 1 pharmaceutics-13-01682-t001:** Techniques to improve solubility and bioavailability and/or dissolution rate of poorly water soluble drugs.

Approach	Technique	Mode of Enhancement	Reference
**Physical**	Micronization or nanoparticles using sizing-down (top down) or building-up (bottom up) technologies	Increase in diffusional surface area	[13,14]
	Inclusion complexes of drugs using cyclodextrins	Increase in solubility via H-bonding	[9,15]
	Self-emulsifying emulsion and microemulsion systems (SEDDS, SMEDDS) using oils, lipids and/or surfactants	Micellar solubilization, reduced efflux and vesicular entrapment	[16,17,18]
	Conversion of crystalline drug into a stabilized amorphous form (solid dispersion, solid solution)	Increase in kinetic solubility and supersaturation	[19,20]
**Chemical**	Salt forms of poorly soluble acidic or basic drugs	pH-partition (via ionization or dissociation)	[21]
	Conversion to inactive but soluble pro-drugs with functional groups such as esters for metabolic conversion to parent drug	Reduced lipophilicity and optimized ADME properties	[22,23]
	Conversion to co-crystals with a conformer	Higher solubility	[24]

**Table 2 pharmaceutics-13-01682-t002:** Review of dissolution studies on amorphous solid dispersion formulations.

Drug (Marketed Product)	Technology (1)	Polymer (2)	Dissolution Approach (3)FDA Recommended Method (  ) vs. Bio-Relevant Approach (∞)	Supporting Analytical Techniques	Conclusion
Reference product of Verapamil: ISOPTIN-SRE, ER tabletsDeveloped formulation(tablets) [93]	HME (ISOPTIN-SRE)Kneading, solvent and co-precipitation method	HPMC/HPC12 SDs prepared:1:1, 1:2 and 1:3 API—polymer ratios with the following polymers: PVPK30, β-cyclodextrin, PEG 6000, HPMCK100M	 Two-phase dissolution:phase 1: 900 mL SGF without enzyme 60 min; phase 2: withdraw and transfer to 900 mL SIF without enzyme, 7 h50 rpm, USP II with wire helix ∞Phosphate buffer pH 1.2, 900 mL, 50 rpm, USP I	DSC, PXRD, SEM, FTIR, supersaturation solubility testing, stability studies	Increased dissolution rates of tablets containing SD with API: PEG6000 ratio of 1:3 in comparison to other formulations and marketed tablets due to decreased particle size, increased wettability and dispersibility of verapamil; Drug–carrier interaction observed; Higher polymer concentration gives faster drug release.
Reference product of Itraconazole:Sporanox cps, and ONMEL tblDeveloped formulation(SD, tablets) [94]	Spray lavering (Sporanox)HME (ONMEL)Solubilization in concentrated aqueous solutions of weak organic acids and drying	HPMCSDs with 2–20% drug load prepared with Glutaric acid	 0.1 N HCl, 900 mL75 rpm, USP II (tbl)SGF without enzyme, 900 mL, 100 rpm, USP II (cps)∞0.1 N HCL, 250 mL75 rpm, USP II (ASD)	DSC, PXRD, ATR-FTIR, pH-solubility studies	Solubility greatly enhanced compared to amorphous form of drug, possible weak drug–acid interactions observed; precipitated as mostly nanoparticles that enable rapid re-dissolution, which might influence absorption.
Reference product of Tacrolimus PROGRAFDeveloped formulation(SD) [95]	Spray drying/fluid bed (PROGRAF)Spray drying via solvent-evaporation method, solvent-wetting method, or surface-attached method – three different processing methodologies	HPMC3SDs prepared:10:80:1 API:HP-β-CD:DOSS ratio	 0.005% HPC in Water with 0.50% SLS adjusted to pH 4.5, 900 mL,100 rpm, USP II (tbl)  HPC solution (1 in 20,000), adjusted to pH 4.5 by phosphoric acid, 900 mL50 rpm, USP II (cps)∞0.005% HPC in Water with 0.50% SLS adjusted to pH 4.5 by phosphoric acid, 500 mL50 rpm, USP II with sinker (SD)	SEM, DSC, PXRD	The solubility and dissolution were significantly improved bySD preparation method compared to drug powder.
Reference product of NifedipineAfeditabDeveloped Tablets [96]	Melt/absorb on carrier Co-precipitation	Poloxamer or PVP12 SDs prepared:1:1, 1:5 and 1:10 API:polymer ratio with all listed polymers:poloxamer, HPMC, PEG 4000 and PEG 6000 [36]; with each API-Polymer ratios of 1:1, 1:5 and 1:10 tested	 0.5% SLS in SGF without enzyme pH 1.2, 900 mL100 rpm, USP II (tbl ER)  SGF without enzyme, 900 mL50 rpm, USP II (cps)∞SGF without enzyme, 900 mL50 rpm, USP XXI (SD, tbl)	DSC, FT-IR	SD tablets prepared with PEG 6000 and poloxamer showed better release profile than marketed products.
Reference product of Griseofulvin (Gris-PEG)Developed formulation(SD) [97]	HME (Gris-PEG)Solvent evaporation technique	PEG 600024 SDs prepared:3:1, 1:1, 1:2 and 1:9 API:polymer ratios with all polymers: PVP, HPMC, and Eudragit L 100, Eudragit E 100, Eudragit S 100, PEG 8000	 4.0% SLS in water, 1000 mL75 rpm, USP II (tbl)  0.54% SLS in water, 1000 mL25 an 50 rpm, USP II (susp)∞Dissolution studies not performed.	PXRD, mDSC, ATR-IR, Raman spectroscopy	Increased polymer concentration leads to lower drug released because drug binds tighter to the concentrated polymers, however SD is more stable.
Reference product of NimodipineNimotopDeveloped formulation (SD) [98]	Spray drying (Nimotop)HME	PEG9 SDs prepared:1:2, 3:7 and 1:9 API:polymer ratio with polymers:HPMC, PVP-VA, Eudragit EPO	 0.5% SDS in water, 900 mL50 rpm, USP II (cps)∞0.05% SLS in acetate buffer pH 4.5, 900 mL75 rpm, ZRS-8G (paddle)	DSC, XRPD, FT-IR, SEM	Eudragit EPO and PVP-VA showed better miscibility than HPMC. Drug–polymer hydrogen bonding was observed.
Reference formulation of Lopinavir & RitonavirKALETRA tablets and capsulesDeveloped formulation (SD) [99]	HME (KALETRA)Solvent granulation process	PVP-VASDs with various API1:API2:PVP-VA ratios	 Tier 1:0.06 M polyoxyethyelene 10 lauryl ether with 10 mM sodium phosphate monobasic (pH 6.8)  Tier 2: same as tier 1 with no more than 1750 USP units/L of pancreatin, 900 mL (cps)50 rpm, USP II  Test 1: 0.06 M decaethyelene glycolmonododecyl ether in water  Test 2: 37.7 g/L of polyoxyethyelene 10-lauryl ether in water (tbl)75 rpm, USP II∞10 mM phosphate buffer pH 6.8, 250 mL and 0.1 N HCl, 250 mL150 rpm, jacketed beaker	XRPD, FT-IR	Molecular mixing of both components into a single amorphous phase negatively impacts ritonavir dissolution performance in comparison with marketed formulation. Amorphous suppression phenomenon observed in pH-shift dissolution method. It is proposed that dissolution of ritonavir from the surface of the particles in acidic media leaves behind a lopinavir-rich surface which acts as a barrier for the remaining ritonavir to dissolve.
Reference product of FenofibrateFenoglideDeveloped formulation (SD) [100]	HME (Fenoglide)Solvent evaporation method	PEG/Poloxamer 1887 SDs prepared:1:1, 1:2 and 1:3 API:polymer ratio with polymers:Carplex 80 and PEG 4000 and 1:5:6 API:polymer ratio with Carplex 80 and PEG 6000 respectively	 25 mM/50 mM/0.75% SLS in water, 1000/1000/900 mL (40 and 120 mg/48 and 145 mg/54 and 160 mg tbl)50/50/75 rpm, USP II  Phosphate buffer w/2% Tween 80 and 0.1% pancreatin pH 6.8, 900 mL75 rpm, USP II (cps)∞Demineralized water, 900 mL50 rpm, USP II	DSC, PXRD, FT-IR, SEM	The most significant improvement of drug dissolution and amorphization was obtained with SD prepared with drug:Carplex:PEG ratio 1:5:6
Reference product of IvacaftorKALYDECODeveloped formulation (SD) [101]	Spray drying (KALYDECO)HME	HPMCAS9 SDs prepared:1:1 API: polymer (Soluplus, HPMC, Copovidone), each pair with three surfactants (SLS, poloxamer, polysorbate 70)	 50 mM sodium phosphate buffer with 0.7% SLS pH 6.8, 900 mL (tbl)65 rpm, USP II with a sinker∞50 mM sodium phosphate buffer pH 6.8, 900 mL65 rpm, USP II	XRPD, DSC, FT-IR	Improved solubilization by improved wetting of drug substance by hydrophilic carriers which represent rich microenvironment formed at the surface of the drug substance and this leads to improved dissolution rate. No defined drug–polymer interaction was observed.
Reference product of PosaconazoleNoxafilDeveloped formulation(SD, tablets) [102]	HME (Noxafil)Spray drying	HPMCAS1 SD prepared:3:1 API:polymer ratio with polymerEudragit L100	 Acid Stage: 0.01 N HCl, 750 mL; Buffer Stage: 50 mM phosphate buffer, pH 6.8 with 0.37% Polysorbate 80 (after 120 min, to the acid stage, add 250 mL of 0.2 M Phosphate Buffer, 1.46% Polysorbate 80) (tbl DR)75 rpm, USP II  0.3% SLS, 900 mL25 rpm, USP II (susp)∞0.01 M HCl with 34 mM NaCl solution and phosphate buffer with SIF powder pH 6.5CTD apparatus	mDSC, PXRD, SEM, in-vivo study	The in-vitro dissolution data underpredicted in-vivo performance, potentially due to higher driving force for precipitation in-vitro versus in-vivo. Including a concentration-sustaining polymer extragranularly to SD but inside tablet was as effective as including it inside the ASD itself.
Reference product of EverolimusCERTICAN and ZORTRESSDeveloped formulation (SD, tablets) [103]	Melt or spray drying (CERTICAN, ZORTRESS)Solvent-wetting and co-precipitation methods	HPMCSDs with various API:polymer ratios with HPMC were prepared	 Water with 0.4% sodium dodecylsulfate, 500 mL (tbl)50 rpm, USP II∞0.4% SLS solution in water and distilled water, dissolution media pH 1.2, pH 4.0 and pH 6.8, 900 mL50 rpm, USP II	XRPD, SEM, particle size analysis, stability and in-vivo studies	The optimized SD consisted of drug:HPMC weight ratio of 1:15. Tablets with SD created with solvent-wetting technique showed identical release rate to that of commercially available product.
Reference formulation of TelaprevirINCIVEKDeveloped formulation (SD) [104]	Spray drying (INCIVEK)Co-milling with polymers	HPMCAS3 SDs prepared:1:1 API:polymer ratio with polymers:PVP-K30, PEG 6000, HPMC	 1% SLS in water, 900 mL (tbl)50 rpm, USP II∞Distilled water, 0.1 M HCl pH 1.2, phosphate buffer pH 6.8, 900 mL100 rpm, ZRC-8D (paddle)	XRPD, DSC, SEM, FT-IR, cytotoxicity evaluation, stability studies	Hydrogen bonding drug–polymer interaction observed. Drug–polymer SD did not affect efficacy of the drug and showed no toxic side effects to normal liver cells. No comparison to reference product shown.
Reference formulation of VemurafenibZELBORAFDeveloped formulation(SD, capsules) [19]	Co-precipitation method(ZELBORAF)Co-precipitation method	HPMCAS3 SDs prepared:2:3 API:polymer ratio withHPMCP, HPMCAS, and Eudragit L 100-55	 1% Hexadecyltrimethylammonium bromide in 0.05 M phosphate buffer pH 6.8, 900 mL (tbl)75 rpm, USP II∞0.05% hexadecyltrimethylammonium bromide in phosphate buffer pH 6.810 mL/min, USP IV ∞FaSSIF, 900 mL75 rpm, USP II [101]	XRPD, DSC, SEM, stability and in-vivo studies	Among used polymers, HPMCAS was found to be the best to prepare stable SD, based on superior physical stability and faster dissolution. No dissolution comparison to reference product shown.
Reference product of RitonavirNORVIR HIVDeveloped formulation(SD) [105]	HME (NORVIR HIV)Solvent evaporation and melt method	PVP-VA4 SDs prepared:1:4 API:polymer ratio polymers Gelucire, sorbitol (with both listed method):	 60 mM polyoxyethyelene 10 lauryl ether, 900 mL (tbl)75 rpm, USP II  0.1 M HCl with 25 mM polyoxyethyelene 10-lauryl ether (cps)50 rpm, USP II∞0.1 M HCl, 900 mLFaSSIF pH 6.5, 500 mL∞FeSSIF pH 5.0, 1000 mL50 rpm, USP II with sinkers	DSC, XRPD, TEM, FT-IR, in-vivo study	Hydrogen bonding was observed in SD resulting in increased drug solubility as compared to pure drug. Maximum dissolution was obtained with FeSSIF media, which confirmed food-related absorption of drugs. No comparison to reference product available.

(1) and (2) Obtained from [36,37,106,107]. (3) Obtained from FDA—recommended Dissolution Methods or USP Dissolution Methods, otherwise cited.

**Table 3 pharmaceutics-13-01682-t003:** Overview of the Methods/Technologies for ASD preparations and limitations of use.

Methods of ASD Preparation	Principle of ASD Conversion	Limitations
**Melting**		
Fusion/melting	The physical mixture of a drug and the carrier (polymer) is heated (above their eutectic point) then melted, then the melt is cooled and solidified. The final solid mass is crushed or milled or sieved.	Not suitable for heat-sensitive materials. Need for the drug-carrier mixture to be homogenous at the heating temperature.
Hot-melt extrusion		
High shear blending generating melting (KinetiSol^®^)		
**Solvent Based**		
Solvent evaporation by rotary evaporator	Dissolving or suspending the drug and the carrier (polymer) in a solvent or a solvent mixture and then evaporating the solvent using different techniques.	All technologies are not amenable for commercial-scale operations. Finding a common solvent that can dissolve both drug and carrier is a challenge. Complete removal of the organic solvent is a key challenge. This method is limited to a few matrices because only a few high molecular weight materials are fiber forming materials [115].
Lyophilization/Freeze drying		
Electrostatic spinning (similar to forming cotton candy)		
Supercritical Fluid Extraction (SCF)		
Spray drying (SD)		
Fluidized bed-coating/layering (FBL)		

**Table 4 pharmaceutics-13-01682-t004:** Summary of the scale-up methodology regarding process parameters.

Feed Properties	
T_feed_	Defined and controlled at lab scale.
C_feed_	Defined at lab scale with minor adjustments as needed.
F_feed_	Calculated from the evaporation rate needed in order to achieve the same vapor concentration at lab scale and pilot scale.
Drying properties.	
T_out_	Measured directly and is kept constant during scale-up.
T_in_	Calculated from the evaporation rate needed in order to achieve the same vapor concentration on lab scale and pilot scale. However, it may not be always possible due to equipment constraints.
F_drying_	Use the nominal flow of the equipment (limited range) to maintain correct loading and velocity in the air disperser, powder transporting ducts, cyclone and filters.
Atomization.	
P_feed_	Adjusted in order to obtain the same droplet size distribution through the different scales.

T_feed_—feed temperature; C_feed_—feed concentration; F_feed_—feed flow; T_out—_outlet temperature; T_in_—inlet temperature; F_drying_—feed flow rate, P_feed_—atomization pressure air.

**Table 5 pharmaceutics-13-01682-t005:** Attributes (needs) required for new ASD polymers to show differentiation from currently available commercial polymers.

Attributes (Needs)	Differentiation Impact	Comments	Differentiation Targets
Higher drug loading	Lower pill or capsule burden that is known to reduce adherence and increase dropout rate.	Typical drug loading for the commercially available polymer is between 10 and 30% of a ASD product. Taking a hypothetical 1 g drug dose with a 20% *w*/*w* drug load, this would equate to 5 g of drug product requiring multiple large capsules or pills.	Drug loading of challenging molecules >40% *wt*/*wt*.
Longer shelf-life, physical and chemical stability	Reduce development and scale-up risks, enabling acceleration of a product to market.	Multiple companies have experienced stability issues at phase II or III clinical trials with ASD that resulted in delays in development and expensive reformulation.	Increased physical and chemical stability under accelerated condition (40 °C and 40 °C/75% RH).
High oral bioavailability	The product requires the solubility advantage offered by an ASD.	Oral bioavailability can be tested in a preclinical species which is known not to have a toxicity issue or extensive high first-pass metabolism which will make it challenging to differentiate between formulations.	Bioavailability in preclinical species is greater than a standard ASD polymer and has acceptable variability.
Enabling continuous manufacturing or reducing manufacturing complexity	Reduced risk during scale-up and/or reduce complexity of manufacturing.	The industry is moving towards green technologies and reducing or eliminating organic solvents. Some compounds have poor solubility in the organic solvents that are acceptable for cGMP ASD manufacturing or have stability issues in these solvents.	Compatible with continuous manufacturing process or compatible with established manufacturing technologies (e.g., SD or HME, while allowing use of low toxicity solvents in process). Without losing solubility advantage and ensuring no toxicity.

**Table 6 pharmaceutics-13-01682-t006:** Commercially available compounds for testing of new ASD polymers.

Compound	Compound ASD Challenges	Reference
lumefantrine	Poor drug loading in ASD Poor chemical and physical stability Poor processing stability Low oral bioavailability	[174]
vemurafenib	Poor drug loading in ASD Poor chemical and physical stability Poor processing stability Low oral bioavailability	[19]
indomethacin	Poor drug loading in ASD Low oral bioavailability	[175]

## Data Availability

The data presented in Figure 10, Figure 11 and Figure 14 were generated by the authors affiliated with Lek Pharmaceuticals as part of development trials.

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
