# Peer review of "Amorphous Solid Dispersions (ASDs): The Influence of Material Properties, Manufacturing Processes and Analytical Technologies in Drug Product Development"

_pharmaceutics, 2021, doi:10.3390/pharmaceutics13101682_

Round 1

Reviewer 1 Report

  • The general structure is nice, but sometimes violated by cross-references to the back and the front. This disturbs the comprehensive flow (- at least it disturbs my comprehensive flow...) 
  • The initial theoretical considerations (Section 2) is very interesting but the relevance and the connection the "application part" is often not clear
  • The Section 2 (theoretical considerations) needs some restructuring to give a reader the chance to have all relevant terms explained before deeper thermodynamics are explained 
  • The figures from the theoretical part support the explanations very well - only the quality of some figures needs improvement
  • Some Headings may be reformulated to better fit the content of the paragraph
  • Section 3 and 4 appear to be better to summarize together and restructure otherwise some redundancies may occur
  • Section 4.3.2 needed more discussion, not only description, with practical suggestions.
  • There are a lot of information collected from different sources, but sometimes a bit more of an evaluation or classification would be desired
  • Especially Section 5 and 6 are interesting and comprehensible
  •  
  • Further detailed comments see in the attached file
  •  

Author Response

The authors of this article are highly appreciate of the high ranking and detailed feedback provided by both reviewers. We consider that these comments will greatly improve the readability and interest of the article and consequently incorporated the majority of the feedback provided.

The publication with changes incorporated has been submit.

Response to Reviewer 1 specific questions

  1. Reviewer 1 comment: “The general structure is nice, but sometimes violated by cross-references to the back and front (-at least it disturbs my comprehensive flow.. ). Response: The authors have made changes to the article to help address this feedback. Additional suggestions to further improve the flow are welcome
  2. Reviewer 1 comment: “The initial theoretical considerations (section 2) is very interesting but the relevance and the connection to the application part is often not clear. Response: The authors have made changes to the theoretical section to simplify and have added statements to the abstract to highlight that the success of this technology is due to the extensive theoretical knowledge which can be applied as part of the product design and optimization. The following statement has been added to the conclusion:

“The reasons for the successful application of ASD include: the significant theoretical understanding that can be directly applied to the design of the product and optimization of the manufacturing process, availability of different polymer stabilizers, alternative novel and commercially viable manufacturing technologies and alternative product presentations (i.e. tablet, capsule, sachet, granules, etc.) that when combined offer a versatile scalable platform that is suitable for different molecules.

  1. Reviewer 1 comment: “The figures from the theoretical part support the explanations very well – only the quality of some figures needs improvements.” The authors agree and have generated high quality figures.
  2. Reviewer 1 comment: “Some heading may be reformulated to better fit the content of the paragraph.” The authors agree and have made changes.
  3. Reviewer 1 comment: “Section 3 and 4 appear to be better to summarize together and restructure otherwise, some redundancies may occur”. The authors agree and have summarized and restructured the sections.
  4. Reviewer 1 comment: “Section 4.3.2 needed more discussion, not only description, with practical suggestions.” The authors consider this section to provide sufficient practical examples. The text contains one example and Table 2 provides a comprehensive list of literature examples with conclusions.
  5. Reviewer 1 comment: “There is a lot of information collected from different sources, but sometimes a bit more of an evaluation or classification would be desired.” The authors appreciate this feedback, but were not clear on how best to address this request and have not made any changes yet but are looking into how to best to classify the contents
  6. Reviewer 1 comment: “Especially section 5 and 6 are interesting and comprehensible”. Author's response: The authors appreciate this feedback.
  7. Reviewer 1 comment: “Further detailed comments see the attached file.” Author's response: The authors greatly appreciate the substantial amount of time of this detailed review. The minor edits, suggestions and changes to figures have mostly been made.

Reviewer 2 Report

I would recommend reference to the FDA's Inactive Ingredient Database (IID) as a more reliable source of information on an excipient rather than GRAS status - see sections 3 and 6.  In section 6 I would ask the authors to consider referencing the FDA's IID and what information this provides rather than GRAS.  Furthermore, use of the more "official" term of "context of use" rather than "proposed use" in the text should also be considered.  In Section 6 is there any value/rationale for using "novel excipients" which would embrace polymers which are clearly the most commonly used materials at this time/

API properties can play a huge part in ASD success and, as well as dose/drug load, maybe a mention of the API crystallization behavior/tendency, solubility characteristics might be worth alluding to? 

Figure 9 - consider adding drug loading to early development.  In my experience if this is not high enough it can stop an ASD project early on

Figure 10 - shows "Kinetisol" and so perhaps this is worthy of a short comment in in Section 3.2 as a emerging "mechanofusion" process?

Author Response

The authors of this article are highly appreciate of the high ranking and detailed feedback provided by both reviewers. We consider that these comments will greatly improve the readability and interest of the article and consequently incorporated the majority of the feedback provided.

The publication with changes incorporated has been submit.

  1. Reviewer 2 comment: “I would recommend reference to the FDA’s Inactive Ingredient Database (IID) as a more reliable source of information in excipient rather than GRAS status – see section 3 and 6. In section 6 I would ask that authors to consider refering the FDA’s IDD and what information this provides rather than GRAS. Furthermore, use of the more “official” term of “context of use” rather than the “proposed use” in the text should also be considered. Authors response: The authors agree that there needs to be a distinction between new and novel excipients. To address this point a new paragraph shown below has been added. We have also changed the wording to include “context of use”.

“The authors define a “new” polymer as an excipient that is already included in the FDA “inactive ingredient data base for approved drug products”, that is in an existing product at or below the daily quantity reported and via the same route of administration (reference: www.accessdata.fda.gov/scripts/cder/iig/index.cfm). New excipients have lower regulatory burden. A “novel” excipient is defined by the authors as an inactive ingredient , or chemically modified inactive ingredient or “Generally recognized as safe” (GRAS) excipient that has not been previously used in a pharmaceutical product via the route of administration. Novel excipients have higher regulatory burden that will be discussed further in this article.”

  1. Reviewer 2 comment: “API properties can play a huge part in ASD success and, as well as dose/drug load, maybe a mention of the API crystallization behavior/tendency, solubility characteristics might be worth alluding to? Author's response: The authors agree with this excellent suggestion and have incorporated a section on this topic.
  2. Reviewer 2 comment: “Figure 9: consider adding drug loading to early development. In my experience, if this is not high enough it can stop an ASD project early on.” Author's response: Drug loading has been included in the figure.
  3. Reviewer 2 Comment: “Shows “Kinetisol” and so perhaps this is worthy of a short comment in section 3.2 as an emerging “Mechanofusion” process?”. Author's response: The authors have incorporated a paragraph on this technology and incorporated it into table 3.

Kind regards
